# Lipid unsaturation promotes BAX and BAK pore activity during apoptosis

Shashank Dadsena [1], Rodrigo Cuevas Arenas [2], Gonçalo Vieira [3], Susanne Brodesser [4], Manuel N. Melo [3] & Ana J. García-Sáez [1,5] ✉

BAX and BAK are proapoptotic members of the BCL2 family that directly mediate mitochondrial outer membrane permeabilition (MOMP), a central step in apoptosis execution. However, the molecular architecture of the mitochondrial apoptotic pore remains a key open question and especially little is known about the contribution of lipids to MOMP. By performing a comparative lipidomics analysis of the proximal membrane environment of BAK isolated in lipid nanodiscs, we find a significant enrichment of unsaturated species nearby BAK and BAX in apoptotic conditions. We then demonstrate that unsaturated lipids promote BAX pore activity in model membranes, isolated mitochondria and cellular systems, which is further supported by molecular dynamics simulations. Accordingly, the fatty acid desaturase FADS2 not only enhances apoptosis sensitivity, but also the activation of the cGAS/ STING pathway downstream mtDNA release. The correlation of FADS2 levels with the sensitization to apoptosis of different lung and kidney cancer cell lines by co-treatment with unsaturated fatty acids supports the relevance of our findings. Altogether, our work provides an insight on how local lipid environment affects BAX and BAK function during apoptosis.

Apoptosis is the best characterized form of regulated cell death and plays a key role in essential biological processes, such as embryo development, tissue homeostasis or the correct functioning of the immune system. Dysregulation of apoptosis has been associated with disease, including neurodegeneration and tumorigenesis, and most chemotherapeutic treatments against cancer rely on killing the tumor cells by triggering apoptosis[1]. Mitochondrial outer membrane permeabilization (MOMP) is a critical event in the mitochondrial pathway of apoptosis. Opening of the apoptotic pore allows the release of apoptotic factors, such as cytochrome c and SMAC, into the cytosol leading to caspase activation and cell death[2]. Given the relevance of apoptosis induction in anticancer strategies, a mechanistic understanding the regulation of the apoptotic pore is of high interest to optimize therapy[3].

MOMP is tightly controlled by the balance between pro- and anti-apoptotic members of the B-cell lymphoma 2 (BCL2) family of proteins[2]. BCL2–associated X protein (BAX) and BCL2 Antagonist/ Killer 1 (BAK) are key effector proteins of this pathway that directly mediate the opening of the apoptotic pore[4,5]. In healthy cells, BAX and BAK exist in inactive, monomeric forms in the cell. BAX is mainly cytosolic due to constitutive retrotranslocation from mitochondria into the cytosol by antiapoptotic BCL-xL, while BAK is mostly tethered to the MOM via its more hydrophobic C-terminal helix[6–8]. During apoptosis, BAX and BAK accumulate at discrete sites at the MOM and undergo a conformational rearrangement into their active forms, which involve extensive membrane insertion[9–12]. The activated proteins then co-assemble into oligomers of a variety of sizes and shapes, including rings, lines, and arcs[13–15]. The mechanism by which BAX and

[1]Institute for Genetics, CECAD Research Center, University of Cologne, Cologne, Germany. [2]Structural Biochemistry, Bijvoet Center for Biomolecular Research, Utrecht University, 3584CG Utrecht, The Netherlands. [3]Instituto de Tecnologia Química e Biológica António Xavier, Universidade Nova de Lisboa, Oeiras, Portugal. [4]Cluster of Excellence Cellular Stress Responses in Aging-associated Diseases (CECAD), Faculty of Medicine and University Hospital of Cologne, University of Cologne, Cologne, Germany. [5]Department of Membrane Dynamics, Max Planck Institute of Biophysics, Frankfurt am Main, Germany. ✉e-mail: ana.garcia@uni-koeln.de

BAK are activated to induce MOMP has been the subject of intense debate. BH3-effector proteins were thought to be indispensable for their activation and for the neutralization of their inhibition by anti-apoptotic family members[12,16–18]. However, the role of the mitochondrial membrane has been recently emphasized in reports proposing that direct contact with the MOM could activate BAX and BAK independently of BH3-only proteins[19–21], with Mg$^{2+}$ acting as a coupling agent between BCL2 proteins and lipids[22]. Furthermore, the lipid composition of mitochondria has been shown to undergo significant changes during apoptosis, which were linked to the regulation of BCL2 proteins and to mitochondrial permeabilization[23–26].

The apoptotic pore is thought to be a toroidal structure whose edge is lined by both proteins and lipids[13,27,28]. Yet beyond BAX and BAK, our knowledge about the molecular composition of the apoptotic pore is rather limited. In previous work, we showed that the dynamin-like protein responsible for mitochondrial division, DRP1, forms supramolecular complexes with activated, mitochondrial BAX during apoptosis[29]. VDACs have also been propose to regulate BAX and BAK action[30–33]. In the case of lipid constituents of the apoptotic pore, very little is understood. Several studies highlight the relevance of lipids for BCL2 protein family function[34–37], with the mitochondria-specific lipid cardiolipin (CL) playing an important role in BAX activity and oligomerization[26,35,38]. However, technical difficulties to handle lipids, together with issues associated with the dynamic nature of the apoptotic pore (including its growth during apoptosis as well as the natural diffusion of lipids within the fluid membrane), hinder our knowledge whether specific lipids line the pore rim and regulate MOMP.

To address this question, here we investigated the local lipid environment of BAK assemblies by extracting them in near-native conditions using lipid nanodics formed by Styrene-Malic acid copolymers (SMA). SMA copolymers, which comprise of hydrophobic styrene and hydrophilic maleic acid moieties, self-assemble with lipids into nanoparticles and solubilize membrane proteins from cellular membranes. Due to the presence of lipids, SMA-extracted membrane proteins present enhanced stability, which makes this system ideal for downstream structural and biophysical studies[39,40]. SMA-copolymers have been successfully used to co-purify lipids surrounding membrane proteins, thus enabling the understanding of the functional interplay of membrane proteins with their annular lipids[41–43].

In this study, by quantitatively comparing the material isolated using mass spectrometry, we find that the lipid environment of BAX/BAK-containing SMALPs become enriched in unsaturated lipid species under apoptotic conditions. In agreement with a role for membrane unsaturation on the apoptotic pore, we observe that unsaturated lipids promoted BAX pore activity in model membrane systems, as well as in isolated organelles and in cells. These results are further supported by the enrichment of unsaturated lipids at the pore rim in Coarse-grain molecular dynamic (CG-MD) simulations. Depletion of the fatty acid desaturase FADS2, a key enzyme in fatty acid poly-unsaturation, decreases apoptosis induction and the activation of the cGAS/STING pathway downstream mtDNA release in cells. Remarkably, the levels of FADS2 in different lung and kidney cancer cells correlate with sensitization to apoptosis by co-treatment with unsaturated fatty acids. Collectively, our results indicate that the lipid environment around BAX/BAK assemblies becomes more unsaturated during apoptosis, thus promoting their membrane permeabilizing activity, which may be relevant for anticancer therapies.

## Results

### Detergent-free solubilization of mitochondrial BAK with its lipid environment

To identify the lipids involved in the formation of the apoptotic pore, we devised an experimental strategy based on the isolation of BAK together with its near-native membrane environment in healthy and apoptotic cells using SMA copolymers. While both BAX and BAK are key components of the apoptotic pore, we chose to use BAK in these experiments because, unlike BAX, it is constitutively associated with the mitochondrial outer membrane[6,44], thus allowing the isolation of BAK enriched SMALPs also from healthy cells as negative control. We selected mEGFP as a tag for affinity isolation of BAK containing SMALPS because it also offers the possibility to track the cellular distribution of the protein during the experiments. We first measured the kinetics of mitochondrial permeabilization in human osteosarcoma U2OS-BAK KO cells stably expressing mEGFP-BAK treated with a combination of BH3-mimetic drugs[45]. We quantified the loss of TMRE staining, associated with a decrease in mitochondrial potential, and correlated it with the rearrangement of the mEGFP-BAK signal from a homogeneous distribution into discrete sites, as proxies for MOMP (Fig. 1A–C). From these experiments, we determined 50 min as the optimal treatment time for substantial MOMP induction, yet before progression to late apoptosis stages. We then isolated crude mitochondria from these cells and solubilized them with 0.5% SMA copolymers (2:1) into SMALPs using the strategy depicted in Fig. 1D. Following the removal of nonsolubilized material, a small fraction of the SMALPs sample was used to check the size and homogeneity of the resulting nanodiscs. Using dynamic light scattering (DLS) analysis and negative transmission electron microscopy (TEM), we calculated that, under our experimental conditions, SMALPs distributed into a narrow population of particles with an average of ~10–12 nm in diameter (Fig. 1E). To enrich for mEGFP-BAK-containing SMALPs, SMA-extracted mitochondrial membranes were then subjected to a single-step affinity purification step using GFP-Trap MA beads. The presence of mEGFP-BAK in the SMALPs eluted from the beads was detected by immuno-blotting using antibody against GFP (Fig. 1F). Since BAK and BAX co-assemble into apoptotic foci, detection of BAX presence in the mEGFP-BAK enriched SMALPs specifically under apoptotic conditions confirmed the validity of our approach, see ref. 15.

### Mitochondrial membranes are enriched in polyunsaturated lipids during apoptosis

In the next step, purified mEGFP-BAK SMALPs were further processed for lipid extraction and subjected to lipidomics analysis by LC-MS/MS (see Methods). As the lipid composition of mitochondria mainly comprises glycerophospholipids, we first focused the analysis of the lipids extracted from mEGFP-BAK-enriched SMALPs from healthy and apoptotic conditions on phosphatidylcholine (PC), phosphatidyletha-nolamine (PE), phosphatidylinositol (PI), phosphatidylserine (PS), phosphatidylglycerol (PG), and phosphatidic acid (PA). We did not detect any significant changes in the phospholipid classes between healthy and apoptotic conditions (Fig. 2A). Interestingly, closer examination of the individual lipid species showed an enrichment in polyunsaturated species from PC and PE at the expense of saturated PC and PE forms in apoptotic mitochondrial membranes compared to untreated samples (Fig. 2B, C). For other, less abundant phospholipid classes (PI, PS, PA, and PG) we could not detect the same trend, although we cannot discard that this is due to the lower levels of these lipids (Fig. 2D–G).

Cardiolipin is a signature phospholipid of mitochondrial membranes and has been reported to play a role in apoptosis[26,35,38]. While it is mainly found in the IMM at a concentration of around 20%, cardiolipin is also present in the MOM at lower levels of around 1-3% of the total lipid content[46–48]. However, the levels of cardiolipin were very low in mEGFP-BAK-enriched SMALPs analyzed by MS (not shown). This raised the question of whether lipid extraction by SMA is equally efficient for all lipids, thus maintaining a composition comparable to that of the original membrane, or whether the SMA-extraction process is preferential for or, excluding, particular lipid classes. To control for this effect, we compared the lipidome of whole isolated mitochondrial membranes from healthy and apoptotic cells that had been solubilized

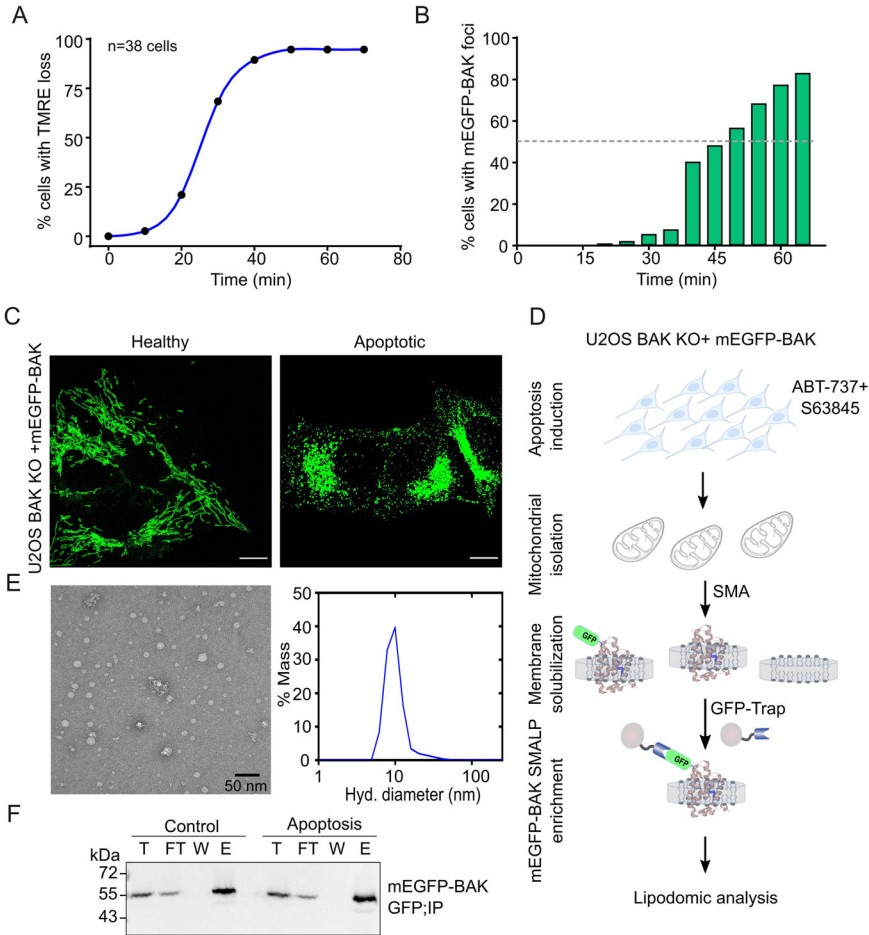

**Fig. 1 | Isolation of GFP-BAK containing SMALPs. A** Loss of TMRE fluorescence due to mitochondrial depolarization is used as a proxy for MOMP progression in the cell population. U2OS BAK KO cells stably transfected with mEGFP-BAK were incubated with 100 nM of TMRE prior to apoptosis induction with ABT-737 (1 µM) and S63845 (1 µM) and measured every 5 min by confocal microscopy.
**B** Redistribution of mEGFP-BAK into foci over time is also used as a proxy for MOMP progression upon apoptosis induction as in (**A**). **C** Representative images of healthy (control) and apoptotic cells expressing mEGFP-BAK (green) from 4 independent experiments. Scale bar, 10 µM. **D** Strategy to isolate SMALPs enriched in mEGFP-BAK from apoptotic cells. U2OS BAK KO cells stably transfected with mEGFP-BAK were incubated with ABT-737 (1 µM) and S63845 (1 µM) for 50 min. Isolated mitochondria from these cells were solubilized with 0.5% SMA co-polymer, and the resulting mEGFP-BAK-containing SMALPs were enriched by affinity purification using GFP-Trap©. "This method adapted from article was published in Molecular Cell, Volume 82, Cosentino et al., The interplay between BAX and BAK tunes apoptotic pore growth to control mitochondrial-DNA-mediated inflammation, Pages 933-949. CC BY license http://creativecommons.org/licenses/by/4.0/ (2022)." **E** Analysis of SMALP size and homogeneity using negative-stain TEM (*left* representative images from 2 independent experiments) and dynamic light scattering (DLS) (*right*). The diameter of the nanoparticles varies between 10 and 12 nm in EM images. Scale bar, 50 nm. A diameter of 12 ± 2 nm was calculated from DLS data. **F** Western blot (WB) analysis of fractions immunoprecipitated from purified mEGFP-BAK-SMALPs from healthy and apoptotic cells. Immunoblots show total input (T), flow through (FT), wash (W) and elution (E) fractions probed with antibodies against GFP. Representative of three independent experiments. Source data are provided as a Source Data file.

into SMALPs prior to lipid extraction, or that had lipids extracted directly without SMA treatment. The lipids extracted from all fractions were again analyzed using LC-MS/MS. Our data demonstrate that the overall phospholipid profiles of mitochondrial membranes with or without SMA treatment were mostly comparable, with no major depletion of the most common lipid types (Supplementary Fig. 1A–D). However, while we were able to detect cardiolipin in the SMALP samples, the amount of SMA-extracted cardiolipin was much lower compared to that detected from bulk mitochondria (without SMA). An enrichment of cardiolipin species in apoptotic total mitochondria was also observed (Supplementary Fig. 1E), in agreement with a previous study[49]. Sphingomyelins and ceramides were not efficiently extracted with SMA either (Supplementary Fig. 1E), although levels of sphingolipid and ceramide species appeared elevated in apoptotic mEGFP-BAK SMALPs compared to control (Supplementary Fig. 1F). Based on this, we concluded that extraction of the lipid environment nearby BAK molecules into SMALPs was adequate to investigate PC, PE, PI, PS and PG changes during apoptosis, but not suitable for the analysis of CL,

sphingomyelins and ceramides. Of note, all lipidomics analysis were performed on lipids extracted directly or from SMALPS from crude mitochondrial fractions, which usually contain cellular impurities (quality test shown in Supplementary Fig. 2A).

Altogether, these experiments showed an increase of unsaturated PC and PE species in SMALPs enriched in mEGFP-BAK from apoptotic cells. Since a similar trend was observed in total mitochondrial extracts, our results suggest that apoptosis induction leads to a general increase in polyunsaturated phospholipid species in mitochondrial membranes that is maintained in the surroundings of the apoptotic pore. These results thus indicate that the lipid environment nearby BAK molecules is altered during apoptosis.

### Membrane unsaturation affects the membrane permeabilizing activity of BAX
Unsaturated fatty acids affect the physical properties of lipid membranes by increasing membrane fluidity and decreasing lateral packing as well as membrane order, whereas saturated fatty acyl chains are

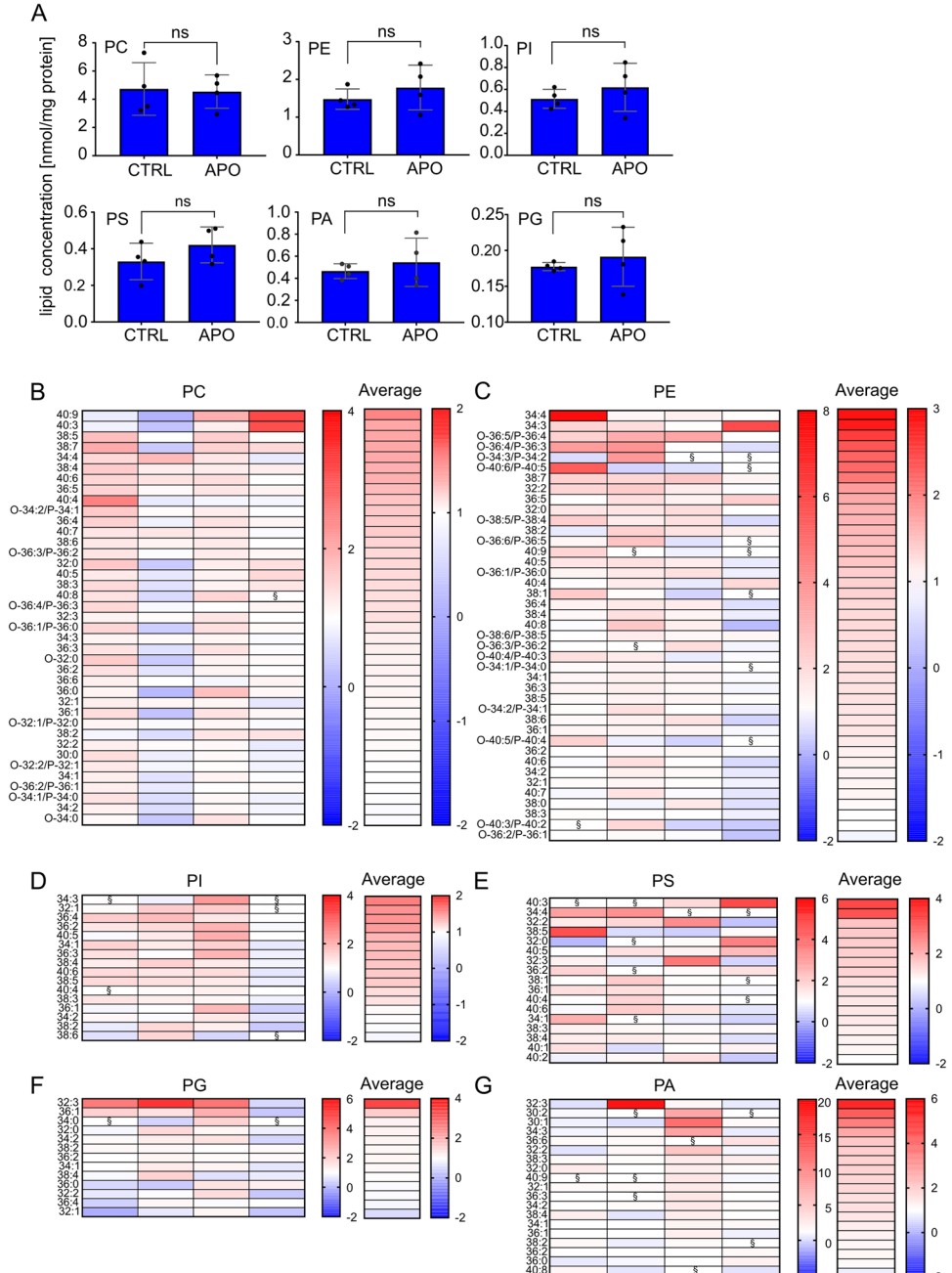

**Fig. 2 | Lipidomic analysis of GFP-BAK containing SMALPs. A** Lipid profile of phospholipids from mEGFP-BAK-SMALPs enriched from mitochondria isolated from healthy and apoptotic cells. The analyses of phospholipid species phosphatidylcholine (PC), phosphatidylethanolamine (PE), phosphatidylinositol (PI), phosphatidylserine (PS), phosphatidylglycerol (PG) and phosphatidic acid (PA) were performed in positive ion mode. Data are presented as means ± S.D., $n = 4$ from independent experiment. $p = 0.2436$; ns, not significant difference by two-tailed paired $t$-test. **B** Heat map of lipid species phosphatidylcholine (PC), (**C**) phosphatidylethanolamine (PE), (**D**) phosphatidylinositol (PI), (**E**) phosphatidylserine (PS), (**F**) phosphatidylglycerol (PG) and (**G**) phosphatidic acid (PA) detected by lipidomics analysis of mEGFP-BAK-SMALPs. Relative fold changes (nmol/mg protein) in apoptotic vs. control samples. $n = 4$, and arranged in descending order of their average values. Source data are provided as a Source Data file.

tightly packed together in the lipid bilayer[50,51]. Since the mechanical properties of lipid bilayers contribute to the energetics of toroidal pores such as the apoptotic pore[13,52–54], we reasoned that the increase in polyunsaturated lipids in mitochondrial membranes during apoptosis might play a functional role on MOMP by regulating the ability of BAX and BAK to form pores. To test this hypothesis, we analyzed the membrane permeabilizing activity of BAX in in vitro reconstituted membrane model systems with varying lipid unsaturation. For these

studies we used BAX instead of BAK, because full-length, wild-type BAX can be produced recombinantly in soluble, monomeric form[53].

We compared the extent of calcein release from LUVs (Large Unilamellar Vesicles) composed of PC:CL (8:2) with varying levels of (un)saturated PC by using fully saturated 1,2-Dipalmitoyl-sn-glycero-3-PC (DPPC), 1-palmitoyl-2-oleoyl-sn-glycero-3-PC (POPC) containing one unsaturation, or 1-Palmitoyl-2-arachidonoyl-sn-glycero-3-PC (PAPC) containing four unsaturations in the acyl chain, in combination

with saturated CL, 1′,3′-bis[1,2-dipalmitoyl-sn-glycero-3-phospho]-glycerol (16:0) (Supplementary Fig. 3A). As shown in Fig. 3A–C, BAX could efficiently permeabilize liposomes composed of mono- and poly-unsaturated lipids in a concentration-dependent manner and exhibited a significantly higher activity in LUVs with polyunsaturated lipids. Remarkably, liposomes with saturated lipids could not be permeabilized even at the higher BAX concentrations tested, likely because the resulting membranes are not in a fluid phase. These results show that the membrane permeabilizing activity of BAX is strongly promoted by unsaturated lipids in the membrane (Fig. 3C).

We then analyzed the pore activity of BAX using Giant Unilamellar Vesicles (GUVs), which allow direct visualization of the permeabilization process as well as the characterization of the pore size, stability, and mechanism (graded *vs* an all-or-none)[53,55]. In this case, we altered the levels of lipid unsaturation by comparing GUVs made of POPC:CL(16:0), containing synthetic saturated forms of cardiolpin with 16 carbon length, or POPC:hCL, containing a natural mixture of cardiolipin from bovine heart with saturated and unsaturated forms of the lipid, both with PC:CL ratio of 8:2. We monitored membrane permeabilization by imaging the entry of a soluble dye (Alexa-488) from the external medium into the lumen of the GUVs initially devoid of dye (Fig. 3D). In agreement with the LUV experiments, we found that BAX presented a higher permeabilizing activity in GUVs containing higher levels of unsaturated lipids compared to GUVs with a lower degree of unsaturated lipids (Fig. 3E). By quantifying the distribution of permeabilization degree in the individual vesicles in the GUV population, we showed in previous studies that BAX follows an all-or-none mechanism of permeabilization, in which the vesicles in the sample exhibit two possible states, impermeable or totally permeabilized[55]. The alternative graded mechanism would imply that the individual GUVs in the sample can exhibit a varying degree of filling. When evaluating here the effect of lipid unsaturation, we found that in both cases BAX-mediated GUV permeabilization continued to follow an all-or-none mechanism, with changes in the levels of non-permeabilized vs. fully permeabilized vesicles (Fig. 3F). Together, these results indicate that the levels of unsaturated lipids affect BAX pore activity but not the permeabilizing mechanism.

We also evaluated whether lipid unsaturation affected BAX pore stability by incubating GUVs with fluorescent probes added at different times. Dye 1(Atto-655) was present in the samples from the beginning. After 1 h, we added dye 2 (Atto-488) and incubated the samples for another hour. We then quantified the number of vesicles permeabilized to dye 1 that were still permeable to dye 2. As expected for stable pores, the majority GUVs filled with Atto-655 were also permeable to Atto633 for all lipid compositions (Fig. 3G), demonstrating that BAX keeps membranes permeabilized for a long period of time independently of lipid unsaturation.

We next examined whether the presence of unsaturated lipids affects BAX pore size. To this aim, we incubated GUVs (POPC:CL(16:0) or POPC:hCL) with cBID/BAX, cyt c-488 (cytochrome c labeled with Alexa 488, 12 kDa) and APC (Allophycocynin, far-red fluorescent protein; 104 kDa) for 1 h (Fig. 3H). Quantitative analysis of the results indicated that BAX effectively induced the internalization of cyt c-488 and APC into GUVs for both lipid compositions (Fig. 3I). Remarkably, internalization of the two fluorescent size markers was higher for the GUVs containing larger amounts of unsaturated lipids. Next, we calculated the degree of filling of individual GUVs in the vesicle population (Fig. 3J, K). For both conditions, internalization of cyt c-488 followed an "all or none" mechanism. In contrast, APC influx followed a graded mechanism in the GUVs containing larger amounts of unsaturated lipids, while the entry of this probe was much lower in GUVs with low lipid unsaturation. These results indicate that the levels of lipid unsaturation affect the size of BAX pores, which can become effectively limiting for larger molecules of about 100 kDa in membranes with lower levels of unsaturated lipids.

## Membrane unsaturation affects the energy pore formation in molecular dynamics simulations

To further validate our lipidomics results, we took the advantage of coarse-grain molecular dynamics (CG-MD) simulations. We first examined whether particular lipid classes or species also exhibited enrichment at the pore rim when the opening of a toroidal, in this case fully lipidic pore, was induced. For this, we first configured an in-silico membrane assembly mimicking the MOM with ~1000 lipids (PC, PE, PI, PG and PS) based on our lipidomics data (Fig. 2). Pores were induced in these systems by imposing position restraints preventing lipid tail particles from accessing a cylindrical region[56]. From the analysis, we observed, for both the cytoplasmic and periplasmic sides, a significant enrichment of PC species in the pore vicinity. This was accompanied by depletion of other lipid species in the cytoplasmic as well as periplasmic leaflets nearby the pore (Fig. 4A and Supplementary Fig. 3C). This is interesting because, while PC is generally considered to lack intrinsic monolayer curvature, a recent MD study reported that PC species with unsaturated lipids have a negative spontaneous curvature[57].

As we found that the degree of unsaturation affects BAX membrane permeabilization activity (Fig. 3B), we argued that membrane unsaturation could also have an impact on the free energy for pore formation. To test this, we used umbrella sampling to obtain the potential of mean force (PMF) profiles and check whether membrane unsaturation shows differences in the pore formation energy in either Martini 3 POPC or PAPC. To be able to obtain energy profiles free of hysteresis artifacts we employed the pore formation coordinate defined by Hub and Awasthi[58] (with pore diameter of 3.0 nm and ζ value of 0.20), rather than the position restraint strategy above. Pores/defects were properly formed across the membranes for both conditions (Fig. 4B, snapshots taken at a pore coordinate of ~0.8). Interestingly, we found a clear increase in energy for pore formation for POPC compared to PAPC from the analysis of the PMF curves (Fig. 4C). These results suggest that PAPC, in spite of its longer tail, contributes to a membrane that is more flexible and more amenable to pore formation, in line with the data obtained in the pore activity assays in reconstituted model membrane systems.

## Loss of poly-unsaturation enzyme FADS2 reduces MOMP and apoptosis sensitivity

Since unsaturated lipids accumulate in mitochondria during apoptosis and promote the membrane permeabilizing activity of BAX in model membranes, we reasoned that reducing the lipid unsaturation in mitochondrial membranes would hinder MOMP and apoptotic cell death. Though mitochondria synthesize several lipids on their own, a continuous supply and exchange of lipids is required for maintaining mitochondrial membranes composition[59]. In humans, dietary PUFAs such as linoleic acids (LA; 18:2, n-6) are converted to Arachidonic acid and ω−3 PUFA by fatty acid desaturase 2 (FADS2) and later incorporated into phospholipids by long-chain fatty acid-CoA ligase-4 (ACSL4)[60]. To test this hypothesis, we evaluated the effect of FADS2 depletion in U2OS cells (Supplementary Fig. 3B) on apoptosis induced by BH3-mimetic drugs (combination treatment of ABT-737 and S63845). As shown in Fig. 5A, B, loss of FADS2 significantly decreased the extent of cell death compared to wild-type cells. In a rescue experiment, feeding cells with unsaturated fatty acids (linoleic acid, LA, 18:2) prior to apoptosis induction led to a significant increase in cell death in both wild-type, as well as FADS2 KO U2OS cells. In contrast, the anti-apoptotic effect of FADS2 depletion was further enhanced in cells fed with saturated fatty acids (stearic acid, 18:0).

To check whether the reduced apoptosis sensitivity of FADS2 KO cells could be attributed to reduced ability of BAX/BAK to permeabilize mitochondria independently of other cellular effects, we measured BAX-induced cyt c release from mitochondria isolated from wild-type

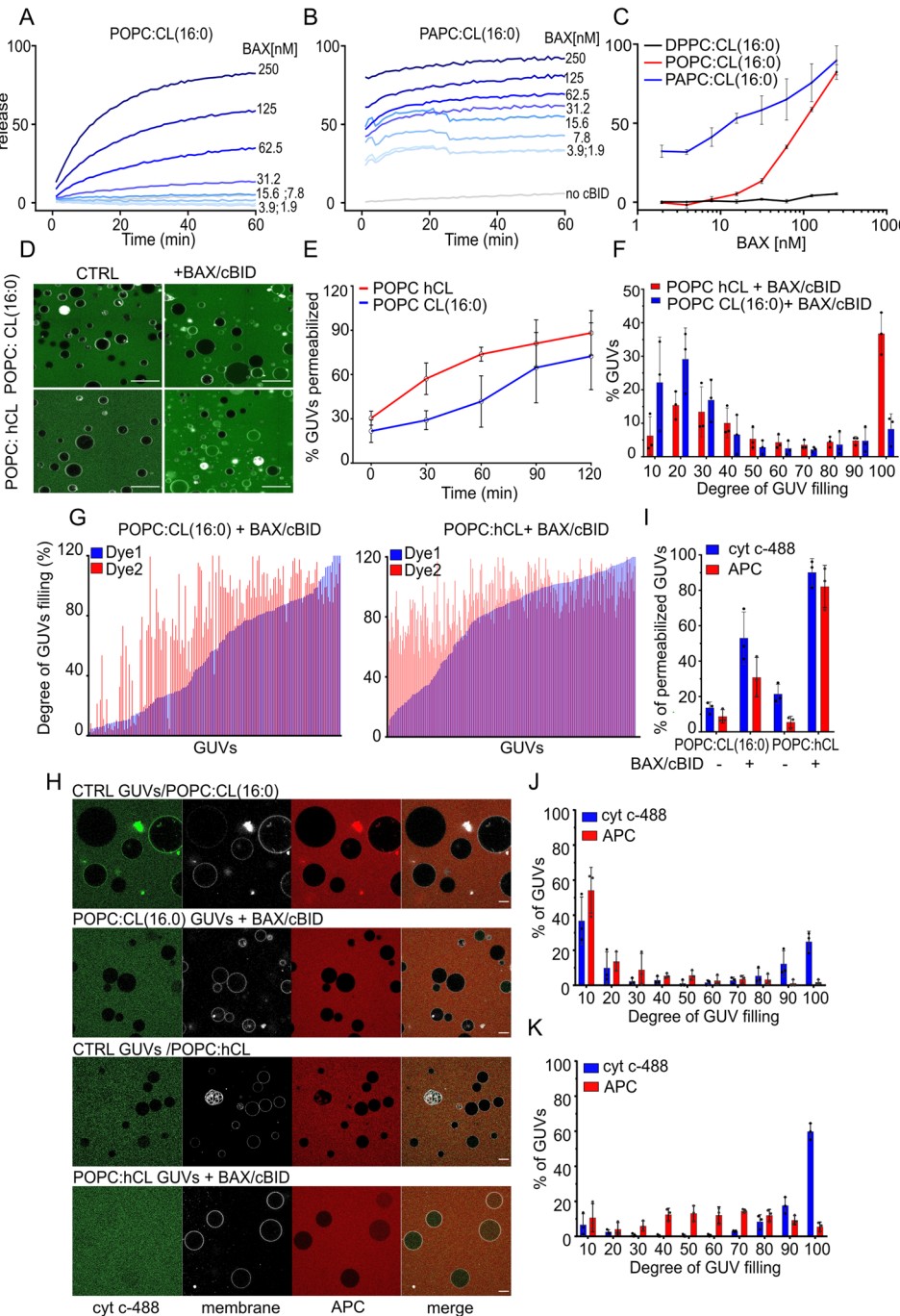

**Fig. 3 | Membrane unsaturation promotes BAX pore-forming activity. A** and (**B**) Kinetics of BAX-induced LUVs permeabilization measured as % carboxyfluorescein release over time. LUVs treated with 10 nM cBID and increasing concentrations of BAX as indicated. LUVs composed of POPC:CL (16:0) (**A**) or PAPC:CL (16:0) (**B**) in 8:2 ratio. Data are presented as means ± S.D.; (POPC, *n* = 4; PAPC, *n* = 5) from independent experiment. **C** Dose-response curve of % carboxyfluorescein release as a function of BAX concentration after 1 h incubation (data from **A**, **B**) for the indicated lipid compositions. Data are presented as means ± S.D.; *n* = 4 (DPPC and POPC); *n* = 5 (PAPC) from independent experiment. **D** Representative images of GUV assay of membrane permeabilization. Internalization of Atto488 dye (green) in the external medium into the lumen of GUVs (membrane shown in grey), initially devoid of dye, indicate vesicle permeabilization. GUV composition was POPC:heart CL (8:2) or PC:CL(16:0) (8:2). Scale bar, 75 μm. **E** % of permeabilized GUVs of the indicated lipid composition at different time points after addition of 100 nM BAX and 20 nM cBID. Data are presented as means ± S.E.; *n* = 3; from independent experiments. **F** Distribution of degree of filling for the individual GUVs after 60 min

incubation with cBID/BAX. Data are presented as means ± S.E.; *n* = 3; from independent experiments. **G** Pore stability assay showing the filling degree of individual GUVs with dye 1 (Atto488, blue lines) and dye 2 (Atto655, red lines) after 2 h incubation with cBID/BAX (*n* = 3; from independent experiments).

**H** Representative images of the GUV assay to estimate pore size. GUVs (gray) in a solution of cyt c-488 (12 kDa, blue) and APC (104 kDa, red) were incubated for 1 h in absence or presence of 20 nm cBID and 100 nm BAX. cyt c-488 and/or APC internalization into the lumen of GUVs indicates vesicle permeabilization. Scale bar, 20 μm. **I** % GUVs permeabilized to cyt c-488 (blue bars) and APC (red bars), (**J**) Distribution of the degree of filling of individual GUVs composed of POPC:CL (16:0) with cyt c-488 (blue lines) and APC (red lines) after treatment with cBID/BAX.

**K** Distribution of the degree of filling of individual GUVs composed of POPC:hCL with cyt c-488 (blue lines) and APC (red lines) after treatment with cBID/BAX. Data are presented as means ± S.E.; *n* = 3 from independent experiments. Source data are provided as a Source Data file.

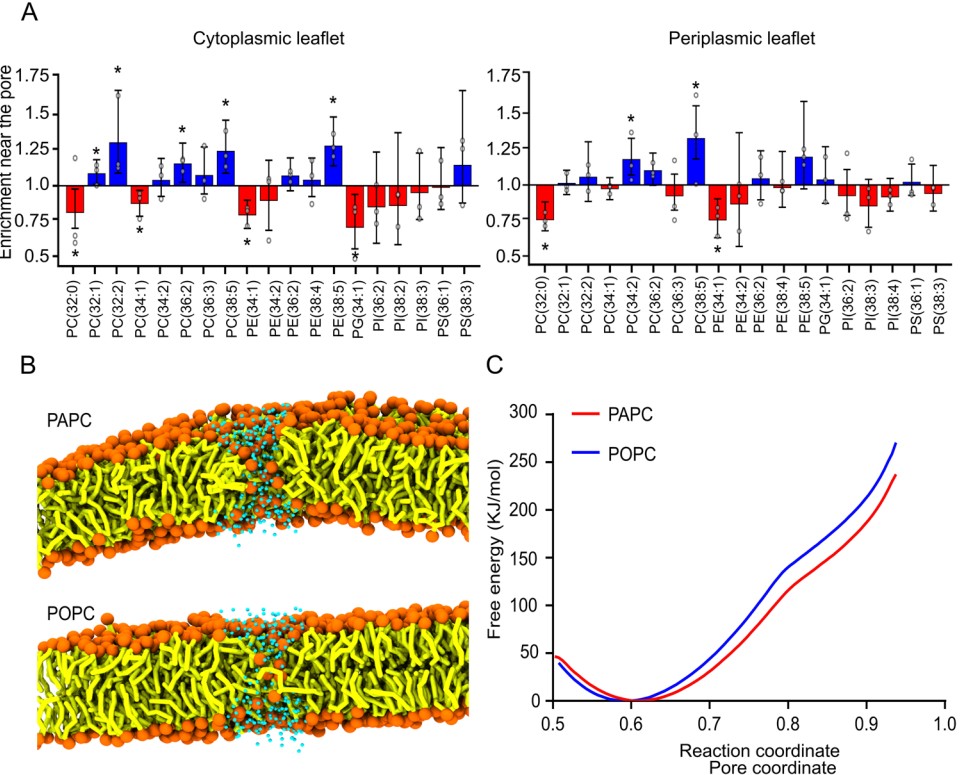

**Fig. 4 | Molecular dynamics simulations of lipid distribution around a toroidal pore. A** Lipid enrichment analysis near the pore. Coarse-grain Martini 3 models on a lipid membrane mimicking the MOM configured based on apoptotic lipidomic data, with pore formation induced by position restraints. Fold change of indicated lipid species around the toroidal pore. 3 points represent the average of each replicates (from the means of triplicate runs analyzed as uncorrelated subtrajectory segments – a minimum 43 segments per replicate for the individual lipid analysis and 62 for the per-headgroup one; error bars indicate the boundaries of a 99% confidence interval; *: *p* values < 0.01 by two-tailed, one-sample, Student's *t*- test; confidence intervals and p-values Bonferroni-corrected for multiple comparisons). **B** Snapshots from an MD simulation after the pore/defect was formed (pore coordinate -0.88) in PAPC or POPC. Phosphate (orange), tails/glycerols (yellow), water beads in the pore vicinity (cyan); the remainder of the solvent and ions are omitted for a clear view, as are choline beads. **C** Potential of mean force (PMF) profile as a function of pore coordinate, for pores formed in PAPC and POPC bilayers and simulated with Martini 3. The error estimated from the weighted histogram analysis (WHAM) is of magnitude between $10^{-1}$–$10^{0}$ kJ/mol and thus not visible in the $10^{2}$ kJ/mol scale of the PMF profiles.

and FADS2 KO MEFs. Remarkably, mitochondria isolated from FADS2 KO cells showed reduced levels of cyt c release compared to wild type (Fig. 5C). In addition, we could increase cyt c release by feeding unsaturated fatty acids to FADS2 KO cells and wild type cells 18 h prior to mitochondrial isolation (Fig. 5C, D). In agreement with this, Supplementary Fig. 4 shows the total fatty acid content of isolated mitochondria of wild-type or FADS2 KO cells supplemented or not with LA analyzed by mass spectrometry. As expected, mitochondria of FADS2 KO cells had less polyunsaturated fatty acids. In addition, LA treatment led to an increase of this fatty acid, which was further metabolized to 18:3, 20:3, and 20:4 species in wild type but not FADS2 KO cells.

In previous work, we found that BAX and BAK assembly determines the growth rate of the apoptotic pore, which then impacts the kinetics of mtDNA release in to the cytosol and the downstream activation of the cGAS/STING pathway under low caspase activity[15]. Since the GUV experiments in Fig. 3 suggested that unsaturated lipids affect the size of BAX pores, we hypothesized that FADS2 depletion could also reduce BAX/BAK pore size also in mitochondria, with consequences for cGAS/STING signaling. Accordingly, we detected a slight delay in STING degradation and TBK1 phosphorylation, accompanied by a reduction in IRF3 phosphorylation in FADS2 KO MEF cells compared to wild type (Fig. 5 E, F). Together, these experiments indicate that reduced mitochondrial membrane unsaturation resulting from FADS2 depletion decreases apoptosis sensitivity, which is linked with a reduced ability of BAX and BAK to induce MOMP. This impacts not only cyt c, but also inflammatory signaling downstream mtDNA release.

Changes in phospholipid metabolism have been described in many cancers[61–63], with FADS2 expression levels altered in some cancer types, such as lung, kidney, and colon cancers[64]. To explore the potential relevance of our findings in cancer, we analyzed the expression level of FADS2 in several cell lines of lung and kidney cancer cells and found that among them, A549 and 786-O cells exhibited robust expression of FADS2 (Fig. 6A) (for reference, expression levels of FADS1 and ACSL4 are shown in Supplementary Fig. 2B). Remarkably, feeding the different cell lines with unsaturated fatty acids prior to apoptosis induction only promoted a significant dose-dependent sensitization to cell death in A549 and 786-O cells expressing high FADS2 (Fig. 6B, C). While correlative, these experiments suggest that the levels of FADS2 expression might be useful markers for the responsiveness of tumors to combination therapies of unsaturated fatty acids with apoptosis inducers.

## Discussion

In this study, we investigated the lipid environment of the apoptotic pore formed by BAX and BAK using a strategy based on lipid nanodiscs. We found an enrichment of polyunsaturated lipid species nearby BAK molecules in apoptotic mitochondria that reflected more global changes in the mitochondrial lipidome and that promoted mitochondrial permeabilization by regulating BAX/BAK pore-forming activity.

The study of the lipid environment contribution to biological processes in membranes has been hampered by technical difficulties to analyze the local lipid composition surrounding the membrane

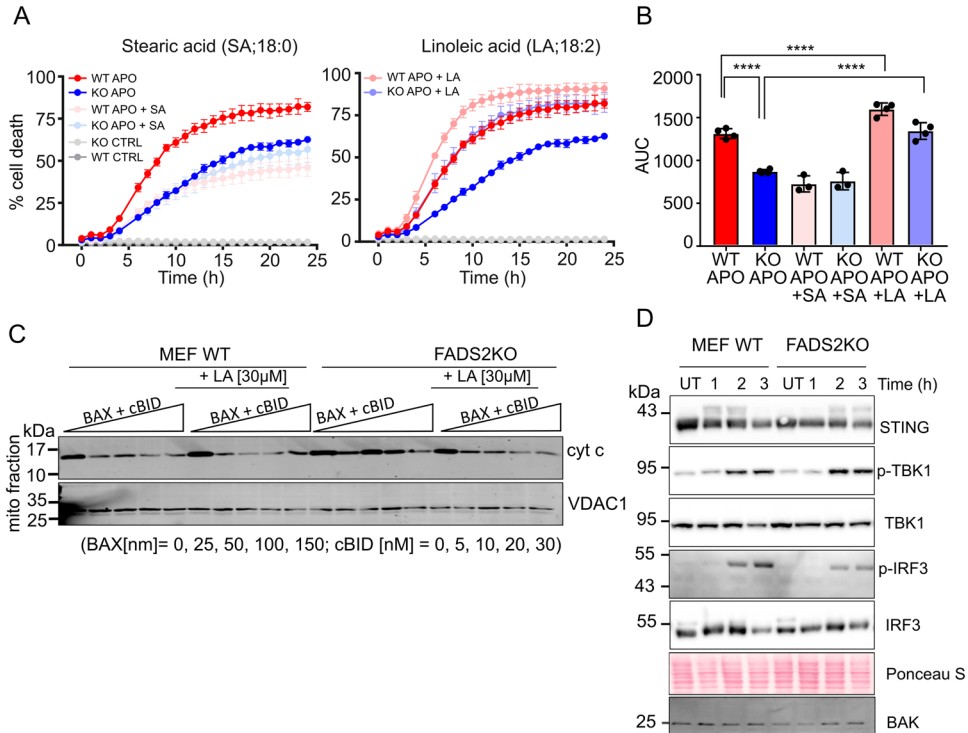

**Fig. 5 | Membrane unsaturation promotes MOMP and apoptotic cell death.**
**A** Kinetics of cell death (measured as %DRAQ7 positive cells) in WT and FADS2 KO U2OS cells. Cells were fed with lipids (stearic acid, linoleic acid or vehicle) for 18 h, and then treated or not with ABT-737 (10 μM) and S63845 (10 μM) for apoptosis induction. Data are presented as means ± S.D..; n = 4 independent experiments. **B** Comparison of the samples in (**A**) by quantification of the area under the curve (AUC). Data are means ± S.D; n = 4; ****, p = 0,0001 by one-way ANOVA corrected for multiple comparisons using Tukey's multiple comparison test) (**C**) WB analysis of cytochrome c release from mitochondria isolated from WT or FADS2 KO MEFs fed with lipids or not for 18 h. Isolated mitochondria were treated with increasing

amounts of BAX and cBID as indicated for 1 h prior to separation of mitochondrial pellets and supernatants. Cytochome c level in the mitochondrial fraction were then detected by immunoblot using anti-cytochrome c antibody. Representative Western blot of n = 3 independent experiments. **D** Activation of the STING pathway downstream of MOMP. WT or FADS2 KO MEFs were treated with ABT-737 (5 μM) and S63845 (5 μM) for the indicated times in the presence of pan-caspase inhibitor. Levels of STING, phosphorylated TBK1, and IRF3 were detected by WB. Actin used a loading control. Representative Western blot of n = 3 independent experiments. Source data are provided as a Source Data file.

proteins of interest. In recent years, the field of lipid nanodiscs has matured to enable mild solubilization and extraction of cellular membranes and their components into macromolecular assemblies that maintain the near-native bilayer structure within discoidal structures of tens of nanometers in diameter[41,65,66]. The SMA-based "detergent-free" extraction of membrane proteins with their native membrane environment enables the quantification of total phospholipids and detailed analysis of the chain lengths and saturations by mass spectrometry[42,67,68]. Here, we took advantage of this approach to investigate the lipid composition of the apoptotic pore. Although the nanodisc size may impose limitations to the size of the BAK supramolecular assemblies that can be extracted, the quantitative lipidome comparison of BAK-enriched SMALP nanodiscs from healthy and apoptotic cells revealed an overall increase in unsaturated lipids, which was more evident for polyunsaturated species of the major phospholipids PC and PE. Given that BAK is a core component of the apoptotic pore, our results indicate that (poly)unsaturated lipids become enriched in the vicinity of the apoptotic pore.

By comparing the lipid composition in total mitochondrial extracts and mitochondrial SMALPs from healthy cells, we obtained an estimation of how accurate lipid extraction with SMALPs reflects the global mitochondrial lipidome under the different conditions. Such analysis revealed that, while most phospholipid species were adequately extracted into SMALPs, both cardiolipins and sphingolipids were not efficiently incorporated into the lipid particles. This unfortunately precludes their analysis using the SMALP strategy, despite the role that has been attributed to these lipids in apoptosis[25,36,69]. Yet, the

increase in cardiolipins and ceramides detected when comparing the total mitochondrial lipidome of apoptotic vs healthy conditions, in line with previous literature[25,70,71], supports the validity of our lipidomic analysis.

Interestingly, the enrichment in unsaturated phospholipids during apoptosis was also reflected in the global mitochondrial lipidome, suggesting that these changes in lipid composition form part of the range of mitochondrial alterations observed during apoptosis[72]. Although we do not know which specific process may be responsible for the increase of unsaturated lipids in apoptotic mitochondria, it might be associated with metabolic changes consequence of cristae remodeling and loss of mitochondrial potential[73,74], and/or with lipid transfer alterations through changes in the contact sites with the endoplasmic reticulum[75,76]. It is important to note that in our experiments we used BH3-mimetics for apoptosis induction, which means that the cause for the increase in unsaturated lipids in apoptotic mitochondria lies downstream of the neutralization of anti-apoptotic BCL2 family members. However, since Venetoclax has been shown to also induce metabolic reprogramming of mitochondria independently of BCL2 inhibition[77], we cannot discard a role for this off-target effect.

We show that the increase in the unsaturation of lipids nearby the apoptotic pore strongly promotes the pore activity of BAX and BAK not only in vitro, but also at the organelle level and in cells, and it is further supported by MD simulations. Indeed, the fact that the contribution of unsaturated lipids is observed in chemically controlled reconstituted systems indicates that their role can be attributed to a direct effect on BAX and BAK, likely via a decrease in the energy for

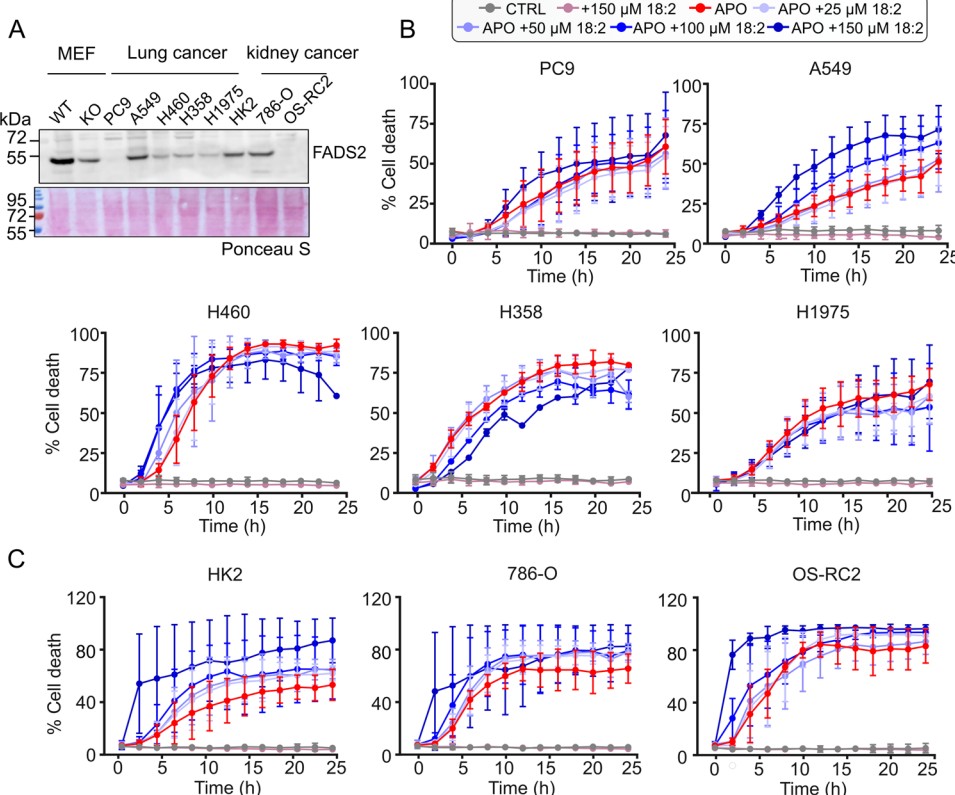

**Fig. 6 | FADS2 expression levels correlate with apoptosis sensitization upon co-treatment of BH3-mimetics with unsaturated fatty acids. A** WB analysis FADS2 expression levels in lung cancer cells (PC9, A549, H460, H358, and H1975), human kidney cells (HK2), and kidney cancer cells (786-O and OC-RC2). **B** Cell death kinetics, measured by % DRAQ7 positive cells, in lung cancer cell lines. Cells were fed with the indicated amounts of unsaturated lipids (linoleic acid or vehicle) for 18 h, then treated or not with ABT-737 (10 μM) and S63845 (10 μM) for apoptosis induction. Data are presented as means ± S.D..; *n* = 3 independent experiments. **C** Cell death kinetics, measured by % DRAQ7 positive cells, in kidney cancer cell lines. Cells were fed with the indicated amounts of unsaturated lipids (linoleic acid or vehicle) for 18 h, then treated or not with ABT-737 (10 μM) and S63845 (10 μM) for apoptosis induction. Data are presented as means ± S.D.; *n* = 3 independent experiments. Source data are provided as a Source Data file.

pore opening by increasing bilayer disorder. Accordingly, changes in lipid composition affect the mechanical properties of the membrane such as fluidity, lateral packing pressure[78,79], membrane bilayer thickness and hydrophobic mismatch tension[80]. Given that the surface properties of the membrane remain largely unaltered by changes in the double bonds of the lipid acyl chains, unsaturated lipids may not affect BAX/BAK membrane binding, but rather act at the levels of membrane insertion, oligomerization and/or line tension at the pore edge. The latter is supported by the reduction in the energy of the pore made of PAPC *vs* POPC detected here using MD simulations. Together, our findings reveal MOMP regulation and pinpoint a mechanism how the lipid environment regulates mitochondrial permeabilization by BAX and BAK with functional consequences for apoptosis execution and cell death.

Our findings are of functional and potentially clinical relevance in the context of tuning therapy responsiveness. We show that the properties of the apoptotic pore can be genetically manipulated by depletion of the fatty acid poly-unsaturation enzyme FADS2, which not only reduces MOMP sensitivity but also the size of the BAX/BAK pores in mitochondria, affecting the downstream activation the cGAS/STING pathway. The effects of lipid (un)saturation on mitochondrial permeabilization and apoptosis sensitivity can be recapitulated by feeding the cells with fatty acids of different unsaturation, which may have consequences for the use of bioactive lipids targeting apoptosis for therapy. Our work provides mechanistic insight into the observed effect of fatty acids on apoptosis sensitivity[81–83] and the specific lipid imbalances associated with certain cancers[84–86]. Along these lines, the

anti-proliferative and pro-apoptotic effects of long-chain poly-unsaturated fatty acids has been associated with reduced risk for the cancer progression[82,87–90]. In this scenario, our data also suggest that the expression levels of FADS2 might serve as biomarker for responsiveness to apoptosis sensitization by fatty acids and support specific dietary regimens in association with anticancer therapies.

In summary, here we report an enrichment of polyunsaturated lipids in apoptotic mitochondria and in the vicinity of the apoptotic pore. We show that increased lipid unsaturation has functional consequences for apoptosis progression by promoting the pore activity of BAK and BAX and thus cell death sensitivity. This not only affects cyt c release for caspase activation, but also inflammatory cGAS/STING signaling in apoptosis. Together, our findings support the relevance of lipid-mediated regulation of BAX/BAK activity and provides mechanistic insight into the pro-apoptotic function of unsaturated lipids that may guide new therapeutic strategies based on bioactive lipids.

## Methods
### Reagents
1,2-Dipalmitoyl-sn-glycero-3-PC (DPPC), 1-Palmitoyl-2-arachidonoyl-sn-glycero-3-PC (PAPC), 1-palmitoyl-2-oleoyl-sn-glycero-3-PC (POPC), 1′,3′-bis[1,2-dipalmitoyl-sn-glycero-3-phospho]-glycerol (16:0 Cardiolipin) and Heart Cardiolipin were purchased from Avanti Polar Lipids. Allo-phycocyanin (APC) was from Biomol and Atto 488, Atto 655, DiI dye were from Thermo Fisher scientific and GFP Trap magnetic agarose beads was from Chromotek. SMA 2:1 (XIRAN) was kind gift from poly-scope. ABT-737, S63845 and Q-VD-OPh were purchased from Hölzel.

Draq7 from Biolegend and 6-Carboxyfluorescein was from Sigma-Aldrich (Germany). Anti-BAK (CST; #5023) anti-BAX (CST; #2772), anti-VDAC1(CST; #4661), anti-Cytochrome C (BD; #556433), anti-FADS1(Proteintech; # 10627-1-AP), anti-FADS2 (AO; #ABIN5963439), anti-ACSL4 (Santacruz; #47997), anti-STING (CST; #50494), IRF3 (CST #4302), Phospho-IRF-3 (CST; #29047), TBK1/NAK (CST #3013), Phospho-TBK1 (CST #5483). anti-Actin (Sigma; #A5316), goat anti-mouse (#31430), goat anti-rabbit (#31460; IB 1:5,000) and donkey anti-goat IgG conjugated to horseradish peroxidase (#pa1-28664; IB 1:5,000) were from Thermo Fischer Scientific.

## Cell culture

Human osteosarcoma U2OS WT, U2OS BAK Ko expressing GFP BAK, U2OS FADS2KO cells, MEF WT and ACSL4 KO cell line were cultured at 37 °C and 5% CO2 in DMEM supplemented with 10% FBS and 1% penicillin/streptomycin (Invitrogen, Germany). All cell lines used in this study were subjected to regular mycoplasma testing using kit (MycoStrip; Invivogen). For lipidomic experiments cells were incubated with 1 μM of ABT-737 and S63845 in the complete media and incubated for 50 min at 37 °C and 5% CO2. FADS2 KO cells in MEF and U2OS cells were generated in the lab by CRISPR/Cas9 method. Linoleic acid or Stearic acid stocks (50 mM) were prepard in ethanol and diluted into culture media before adding them to the cells[91]. Lung cancer cells (PC9, A549, H460, H358 and H1975) were procured from Nieves Peltzer lab (CECAD institute, University zu Köln). Kidney epithelial cells (HK2) and Clear cell Renal cancer cells (786-O and OS-RC-2) were procured from prof. Christian Frezza (CECAD institute, University zu Köln).

## TMRE loss measurement and visualization of BAK foci formation

Cell were incubated with 100 nM of TMRE (Mitochondrial Membrane Potential dye) for 15 min at 37 °C, washed with conditioned media and apoptosis was induced using ABT-737 (1 μM) and S63845 (1 μM). Images were taken every 10 min immediately after treatment in IncuCyte and TMRE signal loss at the single cell level was analyzed using ImageJ. % of cell with TMRE loss was calculated and presented in Fig. 1A. Similarly, to visualize at what time majority of cells show BAK activation and foci formation, apoptosis was induced using ABT-737 (1 μM) and S63845 (1 μM) and images were taken every 5 min immediately after treatment in a SP8 Leica confocal laser microscope. % of cells with activated BAK, detected by means of foci formation, were calculated from total number of cells imaged.

## Crude Mitochondria isolation and membrane solubilization

Mitochondria were isolated from cultured human osteosarcoma cells by mechanical disruption of cells followed by differential centrifugation as previously described[92]. Briefly, cells were harvested by trypsinization, washed in PBS, and then resuspend in isolation buffer (IM;250 mM sucrose, 5 mM Tris, and 2 mM EDTA; pH 7.4 and protease inhibitor cocktail) and mechanically broken using glass homogenizer on ice (30-40 strokes on ice) and total cellular lysates were spin down first to remove nuclei and cell debris at 600 x g for 5 min and later at 10,800 x g for 10 min at 4 °C to get the crude mitochondria. Mitochondrial pellet was washed 2-3 times with isolation buffer to removes other impurities from mitochondria (Supplemetary Fig. 2A). Isolated mitochondria were solubilized using SMA co-polymer. For this, mitochondria either from apoptotic or healthy cells were incubated with 0.5% SMA (2:1) for 45 min at room temperature with gentle rotation. Mitochondrial membrane was spun down for 100,000 x g for 40 min to separate solubilized SMALP from the insolubilized membrane. Next, the Size of SMALP were analyzed by Dynamic Light Scattering (DLS). For DLS measurements, 15 μl of sample was added to a quartz cuvette which had been thoroughly cleaned with Milli-Q H2O. The cuvette was placed in DynaPro NanoStar (Wyatt Technology corporation, USA) and the sample was analyzed using 10 runs with 10 second acquisition time.

This helps to determine the mass distribution of the sample as well as the estimated size of the particles. The distance distribution is shown on a log scale. The size of SMALP as well as the homogeneity with in the sample were also checked by Negative Transmission Electron Microscopy (TEM). For this the diluted SMALPs were placed onto a glow-discharged copper grid (Electron Microscopy Sciences) coated with a layer of thin carbon, washed twice with water, stained with 2% uranyl acetate for 5 min and then air-dried. The grids were imaged on a JEOL JEM2100PLUS electron microscope and recorded with a GATAN One-View camera (CECAD imaging Facility).

## Sample preparation and lipidomic analysis

mEGFP-BAK-SMALPs were affinity purified from total solubilized mitochondrial membrane fraction (SMALP). For this total SMALP were incubated with 25 μl of GFP-trap MA beads for 90 min with slow rotation in cold room. Beads were washed 2 times with 100 μl of Tris buffer (50 mM Tris 150 mM NaCl pH 8), and finally resuspend in 100 ul of Tris buffer. Small aliquot of unbound and wash fractions was used to analyze the purification quality.

## Mass spectrometry analysis of glycerophospholipids

Lipids from isolated mitochondria treated with or without SMA were extracted using a procedure previously described[93] with some modifications: 30-100 μl of sample were brought to a volume of 200 μl with 155 mM ammonium carbonate buffer. Lipids were extracted by adding 990 μl of chloroform/methanol 17:1 (v/v) and internal standards (125 pmol PC 17:0-20:4, 138 pmol PE 17:0-20:4, 118 pmol PI 17:0-20:4, 118 pmol PS 17:0-20:4, 61 pmol PG 17:0/20:4, 72 pmol PA 17:0/20:4, 10 μl Cardiolipin Mix I; Avanti Polar Lipids), followed by shaking at 900 rpm/min in a ThermoMixer (Eppendorf) at 20 °C for 30 min. After centrifugation (12,000 x g, 5 min, 4 °C), the lower (organic) phase was transferred to a new tube, and the upper phase was extracted again with 990 mL chloroform/methanol 2:1 (v/v). The combined organic phases were dried under a stream of nitrogen. The residues were resolved in 200 μl of methanol.

PC, PE, PI, PS, PG, and PA species were analyzed by Nano-Electrospray Ionization Tandem Mass Spectrometry (Nano-ESI-MS/MS) with direct infusion of the lipid extract (Shotgun Lipidomics) as previously described[94]. Endogenous glycerophospolipids were quantified by referring their peak areas to those of the internal standards. The calculated glycerophospolipid amounts were normalized to the protein content of the mitochondria suspension.

Cardiolipin (CL) species were analyzed by Liquid Chromatography coupled to Electrospray Ionization Tandem Mass Spectrometry (LC-ESI-MS/MS): 10 μl of the lipid extract in methanol were loaded onto an Acquity BEH Shield RP18 column (100 mm × 2.1 mm ID, 1.7 μm particle size; Waters), and CL species were detected using a QTRAP 6500 triple quadrupole/linear ion trap mass spectrometer (SCIEX). The LC (Nexera X2 UHPLC System, Shimadzu) was operated at 50 °C and at a flow rate of 0.4 ml/min with a mobile phase of 10 mM ammonium formate in acetonitrile/water 60:40 (v/v) (solvent A) and 10 mM ammonium formate in isopropanol/acetonitrile 90:10 (v/v) (solvent B). CL species were eluted with the following gradient: initial, 30 % B; 0.5 min, 30 % B; 4.5 min, 68 % B; 20.5 min, 75 % B; 21 min, 97 % B; 24 min, 97 % B; 24.5 min, 30 % B; 28 min, 30 % B[95].

CL species were monitored in the positive ion mode using the following Multiple Reaction Monitoring (MRM) transitions: CL 68:4, m/z 1418.9 to 575.4; CL 70:4, m/z 1446.9 to 575.4; CL 72:4 m/z 1475.0 to 603.4; CL 61:1 (internal standard), m/z 1326.9 to 535.4. For all MRM transitions the values for declustering potential, entrance potential, collision energy, and cell exit potential were 140 V, 10 V, 45 V, and 7 V, respectively[96]. The instrument settings for nebulizer gas (Gas 1), turbo gas (Gas 2), curtain gas, and collision gas were 50 psi, 50 psi, 40 psi, and medium, respectively. The Turbo V ESI source temperature was 500 °C, and the ionspray voltage was 4.5 kV.

The LC chromatogram peaks of the endogenous CL species and the internal standard CL 61:1 were integrated using the MultiQuant 3.0.2 software (SCIEX). Endogenous CL species were quantified by normalizing their peak areas to those of the internal standard. The normalized peak areas were then normalized to the protein content of the mitochondria suspension.

## Mass spectrometry analysis of ceramides and sphingomyelins

For the analysis of ceramides and sphingomyelins in isolated mitochondria without and after SMA treatment, lipids were extracted as described above in the presence of 127 pmol ceramide 12:0 and 124 pmol sphingomyelin 12:0 (internal standards, Avanti Polar Lipids). The dried extracts were resolved in 100 μL of Milli-Q water and 750 μL of chloroform/methanol 1:2 (v/v). Alkaline hydrolysis of glycerolipids and LC-ESI-MS/MS analysis of ceramides and sphingomyelins were conducted as previously published[97,98].

## Mass spectrometry analysis of fatty acids

To 100 μl of a suspension of isolated mitochondria in PBS, 500 μl of methanol, 250 μl of chloroform, and 0.5 μg palmitic-d31 acid (Sigma-Aldrich) as internal standard were added. The mixture was sonicated for 5 min, and lipids were extracted in a shaking bath at 48 °C for 1 h. Glycerolipids were degraded by alkaline hydrolysis adding 75 μl of 1 M potassium hydroxide in methanol. After 5 min of sonication, the extract was incubated for 1.5 h at 37 °C, and then neutralized with 6 μl of glacial acetic acid. 2 ml of chloroform and 4 ml of water were added to the extract which was vortexed vigorously for 30 sec and then centrifuged (4,000 × g, 5 min, 4 °C) to separate layers. The lower (organic) phase was transferred to a new tube, and the upper phase extracted with additional 2 ml of chloroform. The combined organic phases were dried under a stream of nitrogen. The residues were resolved in 200 μl of acetonitrile/water 2:1 (v/v) and sonicated for 5 min. After centrifugation (12,000 × g, 20 min, 4 °C), 40 μl of the clear supernatants were transferred to autoinjector vials.

Fatty acid levels were determined by LC-ESI-MS/MS using a procedure previously described[99] with a few modifications: 10 μl of sample were loaded onto a Core-Shell Kinetex Biphenyl column (100 mm × 3.0 mm ID, 2.6 μm particle size, 100 Å pore size; Phenomenex), and fatty acids were detected using a QTRAP 6500 triple quadrupole/linear ion trap mass spectrometer (SCIEX). The LC (Nexera X2 UHPLC System, Shimadzu) was operated at 40 °C and at a flow rate of 0.7 ml/min with a mobile phase of 5 mM ammonium acetate and 0.012 % acetic acid in water (solvent A) and acetonitrile/isopropanol 80:20 (v/v) (solvent B). Fatty acids were eluted with the following gradient: initial, 55 % B; 4 min, 95 % B; 7 min, 95 % B; 7.1 min, 55 % B; 10 min, 55 % B.

Fatty acids were monitored in the negative ion mode using "pseudo" Multiple Reaction Monitoring (MRM) transitions[99]. The instrument settings for nebulizer gas (Gas 1), turbo gas (Gas 2), curtain gas, and collision gas were 60 psi, 90 psi, 40 psi, and medium, respectively. The Turbo V ESI source temperature was 650 °C, and the ionspray voltage was −4 kV.

The LC chromatogram peaks of the endogenous fatty acids and the internal standard palmitic-d31 acid were integrated using the MultiQuant 3.0.2 software (SCIEX). Endogenous fatty acids were quantified by normalizing their peak areas to those of the internal standard. The normalized peak areas were then normalized to the protein content of the mitochondria suspension.

The minimal metadata set for Cardiolipin, Ceramide and sphingomyelins, Fatty acids and GPLs are provided as Supplementary Data 1, Supplementary Data 2, Supplementary Data 3, and Supplementary Data 4, respectively.

## Liposome permeabilization assay

Lipids were purchased from Avanti Polar Lipids, and were dissolved in chloroform and mixed with the desired molar ratios. Chloroform was then evaporated under vacuum for 3 h. Large unilamellar vesicles (LUVs) were prepared using the extrusion method as described before[52,100]. Briefly, the lipid film (PC:CL; 8:2) was hydrated with 80 mM solution of the fluorescent dye carboxyfluoroscein, pH 7 for a final lipid concentration of 5 mg/mL followed by five cycles of freezing and thawing. The lipid solution was then extruded through a polycarbonate membrane with a pore size of 100 nm using glass syringes. Carboxyfluoroscein-loaded LUVs were separated from free carboxyfluoroscein using Sephadex-G50 beads and the lipid concentration was adjusted to 100 μM. LUVs were incubated with serial dilutions of the recombinant proteins in a 96-well plate and carboxyfluoroscein release was monitored by fluorescence emission at 520 nm with excitation at 490 nm for 1 h using a microplate reader (Enspire, PerkinElmer). 0.1% Triton X-100 was used as a positive control (100 % carboxyfluoroscein release) and the percentage of carboxyfluoroscein release was calculated as follows:

$$\% \, release = 100 \, \frac{(Ff - Fi)}{(Fm - Fi)}$$

where $F_f$ is fluorescence measured in the sample at particular time point, $F_i$ is the fluorescence in the negative sample (liposome alone), and $F_m$ is the maximal value upon addition of Triton X-100. Spontaneous release of carboxyfluoroscein was found to be negligible in all cases.

## GUV permeabilization assay

GUVs were produced by electro-formation and the experiments were done as described in ref. 55. Briefly, 3 μl of a 2 mg/ml PCPC:CL (8:2) lipid mixture solution, doped with the lipophilic membrane dye DiI (Thermo Fisher) and dissolved in chloroform, were spread on each platinum electrode of an electro-formation chamber and immersed in 300 mM sucrose. Electro- formation proceeded for 2 h at 10 Hz, followed by 30 min at 2 Hz. 50-100 μl of the GUVs suspension was added to a solution of PBS buffer mixed with the proteins (100 nM BAX + 20 nM cBid) in Lab-Tek 8-well chamber slides (NUNC) to a final volume of 300 μl. To measure the membrane permeabilization activity of BAX, Atto-488 free dye was added to the solution and images were taken at different time interval. In pore stability experiment, GUVs and cBid/Bax protein were incubated first with Dye1(Atto-655 dye) for 1 h, then with Dye2 (Atto-488 dye) in the same solution for another 1 h and then images were taken. For pore size experiment GUV and cBid/Bax protein were incubated with cyt c-al488 (-12 kDa) and APC (100 kDa) and images were taken after 1 h. The percentage of GUVs internalizing the fluorescent probes was determined by the GUV detector software described in ref. 101

## IncuCyte cell death measurements

Cell death assay were performed using IncuCyte S3 (Sartorius) at 37 °C 5% $CO_2$. For this, cells were seeded with low lipid media (DMEM + delipidated serum) in a 96-well plate and treated with either vehicle or with saturated and unsaturated lipids for 16 h. Cells were stained with Draq7 (1:3,000 dilution) in low lipid media and apoptosis was induced using ABT-737 (10 μM) and S63845 (10 μM). After treatment, four images per well were acquired every 1 h for 24 h and analyzed using IncuCyte basic analysis software module. Cell death was measured by the incorporation of Draq7 to the dead cells. Data were collected as count of Draq 7 positive cells per total number of cells in each condition and normalized taking as 100% the total count of cells in each condition.

## Cytochrome c release assay

In order to measure release of cytochrome c from isolated mitochondria, 30-40 × 10^6 MEF WT and ACSL4KO cells were seeded in low lipid media with either vehicle or unsaturated lipids for 16 h. Crude

mitochondria isolation was performed as mentioned before. 50 μg of isolated mitochondria were incubated with BAX (50-100 nM) and cBid (10-20 nM) for 30 min at 30 °C. Mitochondria were pelleted by centrifugation at 14,000 g for 10 min at 4 °C. Both, pellet as well as the supernatant fractions were cooked with laemmli buffer (2-5x) and subjected to SDS PAGE gel. Cyt c level in the mitochondria were detected by immunoblot using Odyssey Infrared Fluorescent Imaging system.

## Western blotting

For affinity purification quality analysis, 10 μl of beads (purified material) were cooked with 2x laemmli buffer at 95 °C for 10 min and run on 12% SDS-PAGE gel and transferred onto nitrocellulose membrane using the Turboblot (BioRad). Blots were blocked with 5% milk in TBST and incubated overnight at 4 °C with the primary antibody (1:1000 dilution), probed with secondary antibodies (1:10,000) and detected using ECL reagent (BioTool). For cytochrome c detection isolated mitochondria and supernatant fraction were directly cooked with 2-5x of laemmli buffer at 95 °C for 5 min and run on SDS-PAGE and transfer to membrane as directed above. Membranes were incubated with mouse anti-cyto c ab and IRDye® 800CW anti-mouse secondary ab and detected using Odyssey® Imagers (LI-COR Biosciences). Detection of other proteins by western blot was performed slightly different than stated above. For this, cells were lysed in triton buffer (150 mM NaCl, 1% Triton, % SDS, 2 mM EDTA and 10 mM Tris, pH 7.4) with protease inhibitors. Protein concentration was determined by Bradford protein assay (Bio-Rad) according to the manufacturers protocol. Equal amounts of protein (60 μg) were loaded on SDS-PAGE gel, transferred to nitrocellulose membrane and detected by western blot using respective antibodies.

## Molecular dynamics simulations

**System setup.** CG MD simulations were performed using the Martini 3 model. For all the used compositions, the insane.py[102] script was used to assemble $20 \times 20$ nm² membranes, with around 1130 lipids and at a hydration of 25 regular Martini waters per lipid. System charge neutralization and 0.15 M ionic strength were setup by adding appropriate numbers of $Na^+$ and $Cl^-$ ions.

The lipid topologies released with the Martini 3 model were used. Any missing head-tail combinations were built following the Martini building-block approach. Different tail lengths or degrees of lipid unsaturation were built analogously to existing tails.

Mitochondrial outer membranes were modeled with asymmetric lipid distributions as follows: lipidomics assays were used to inform the overall lipid types and their proportions. These were then divided into cytoplasmic/intermembrane-space leaflets assuming the proportions in Fig. 1B of Horvath & Daum 2013[59] (30/70, 30/70, 77/23, and 55/45 for PS, PI, PE, and PC, respectively; information on PG, which is a minority component, was not available and 50/50 distribution was assumed), grouping tail types to arrive to at least 5 molecules per lipid species per leaflet. In order to avoid introducing tension due to area-per-lipid mismatch between leaflets, each composition's area-per-lipid was determined from symmetric simulations. Leaflets were then juxtaposed, after proportionally removing enough lipids from the one with highest area-per-lipid, so that expected areas matched.

**Simulation conditions.** The GROMACS 2020 simulation package[103] was used. A Verlet neighbor list scheme was employed, at a 20-fs timestep. Nonbonded interactions followed typical Martini settings[104]: Van der Waals and Coulombic potentials were cut-off at 1.1 nm; Coulombic interactions were computed with a reaction-field scheme, with a dielectric constant of 15. Systems were thermostatted to 300 K by the V-rescale method[105] with a 1 ps coupling time. Pressure was kept at 1 bar by a Parrinello–Rahman barostat[106] coupled semi-isotropically (independently in the xy and z directions) with a 12 ps coupling time. After

system assembly and energy-minimization, but prior to production runs, a 2 ns pressure/temperature equilibration was run for all systems, using the more robust Berendsen barostat[107] with a relaxation time of 3 ps.

**Pore formation.** Pores were induced in assembled membranes in two fashions: in the first, used to assess mitochondrial OMM lipid segregation, pores were formed by imposing a harmonically repulsive potential against all first tail particles, radially along the z-axis and crossing the membrane at the center of its xy dimensions; the resulting cylindrically shaped potential had a 3 nm radius and a force constant of 500 kJ/mol/nm2. These simulations were run in triplicate for each condition, for a minimum 8.3 μs. Pore vicinity, for the purpose of calculating lipid enrichment/depletion was defined as a 4-nm cylindrical zone centered on the pore. Any lipid with a particle inside this cylinder was considered to be pore-associated.

In the second fashion, pores were formed using the method developed by Hub and Awasthi[58], implemented in the Plumed software, version 2.5[108]. Briefly, a collective variable reflecting pore formation was defined as the fraction of slices of a membrane-spanning cylinder where water, ions or headgroup particles were present. The cylinder was setup with a 3.0 nm diameter, a height of 8 nm (40 slices each 0.2 nm thick) and a zeta parameter (related to the continuous counting of slice occupancy[108] of 0.2. This method allowed the imposition of potentials restraining the resulting collective pore variable, and hence the determination of free-energy profiles of pore formation. This was accomplished, for each condition, in an umbrella-sampling setup of 45 windows, run for 100 ns each, except for selected windows that were extended beyond 600 ns each due to longer equilibration times – presumably due to proximity to a transition state. Window sampling data is available in Supplementary Fig. 3. System compositions and simulation times are discriminated in Supplementary Table 1.

**Analysis.** Analysis and visualization made extensive use of the MDAnalysis[109], NumPy[110] and matplotlib[111] Python packages, as well as the VMD 1.9.3 molecular visualization software[112]. Pore enrichments were analyzed from the three replicate trajectories, divided into time-uncorrelated segments (a minimum 43 segments per replicate for the individual lipid analysis and 62 for the per-headgroup one); time correlation was estimated as the largest integrated autocorrelated time of the amount of each lipid species near the pore; this corresponded to a maximum autocorrelation time of 190.0 ns for the individual lipid analysis and of 134.5 ns for the per-headgroup one; these trajectory lengths were also truncated from the beginning of each replicate trajectory, to ensure decorrelation from starting configurations. Significance and confidence intervals were estimated with a two-tailed, one-sample, Student's $t$-test, with Bonferroni correction for the multiple comparisons. Umbrella sampling analysis, including error estimation, was done with Alan Grossfield's Weighted Histogram Analysis Method (WHAM) implementation, version 2.0.11_ [http://membrane.urmc.rochester.edu/wordpress/?page_id=126].

Reliability and reproducibility checklist for molecular dynamics simulations are listed in Supplementary Table 2.

## STING pathway analysis

Cells were treated with ABT-737 (10 μM) and S63845 (10 μM) in presence of pan-caspase inhibitor (20 μM) for 1, 2, and 3 h. After treatment, cells were lysed in RIPA buffer, and equal proteins were loaded on SDS-PAGE. Level of STING, normal, and phosphorylated forms of TBK1 and IRF3 were detected in the western blot.

## Statistical analysis

The data requiring quantification and statistical analysis were presented on the basis of at least three independent experiments. Statistical analysis was performed with Prism 7 software (GraphPad). Data

were shown as means ± SD. Differences between experimental groups were assessed for significance using a two-tailed unpaired Student's *t* test, ***$P < 0.001$. For more than two groups comparison, we assessed one-way ANOVA and corrected for multiple comparisons using Tukey's multiple comparison test).

### Reporting summary

Further information on research design is available in the Nature Portfolio Reporting Summary linked to this article.

## Data availability

Data supporting the findings of this manuscript are available from the corresponding authors upon reasonable request. A Reporting Summary for this Article is available as a Supplementary Information file. The source data underlying Figs. 1, 2, 3, 4, 5, and 6 are provided as a Source Data file. Full scans of immunoblots are provided in the Source data file. The lipidomics data used in this manuscript is available at under (https://www.metabolomicsworkbench.org/data/DRCCMetadata.php?Mode=Study&StudyID=ST003114). Source data are provided with this paper.

## Code availability

Computational dataset, custom code for simulation analysis as well as used models are available at https://doi.org/10.5281/zenodo.11241824.

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

## Acknowledgements

This work was supported by European Research Council (ERC) under the European Union's Horizon 2020 research and innovation program (Grant agreement No. 817758), by the Deutsche Forschungsgemeinschaft (DFG, German Research Foundation), SFB1218, project no. 269925409 and by CANTAR research network "CANcer TARgeting", Ministry of Culture and Science of the State of Northrhine Westphalia, Germany and by FCT – Fundação para a Ciência e a Tecnologia, I.P., through MOSTMICRO-ITQB R&D Unit (doi:10.54499/UIDB/04612/2020; doi:10.54499/UIDP/04612/2020), LS4FUTURE Associated Laboratory (doi:10.54499/LA/P/0087/2020) and CEECIND fellowship to M.N.M. (doi:10.54499/CEECIND/04124/2017/CP1428/CT0008). Special thanks to Prof. Christian Frezza and Dr. Nieves Peltzer for providing kidney and lung cancer cell lines. We thank Dr. R. Shalaby, Dr. H. Flores, Dr. L. Pedrera Puentes and Prof. A. Villunger for their technical input and stimulating discussions

## Author contributions

S.D. and A.J.G.S. conceptualized the study and wrote the manuscript through the contributions of all authors. S.D. designed and performed all the experiments with cellular system and model membranes. S.B. performed mass spectrometry for all the lipidomic samples and analyzed the data. M.N.M. designed, performed, and analyzed the computer simulation studies, with critical input from G.V. A.J.G.S. conceived the project with input from R.C.A.

## Funding

## Competing interests

The authors declare no competing interests.
