## [Peer Review File · Nature Communications]

Lipid unsaturation promotes BAX and BAK pore activity during apoptosisREVIEWER COMMENTS

Reviewer #1 (Remarks to the Author):

The manuscript by Dadsena et al. focuses on the role of lipid unsaturation near BAX and BAK for mitochondrial outer membrane permeabilization (MOMP). Combining analysis of SMA stabilized BAK-containing lipid nanodiscs with molecular dynamics (MD) simulations they show that BAX/BAK prefer unsaturated lipids in their nano-environment and that this lipid environment may promote membrane permeabilization. Overall a very nice study!

The following points need some clarification:

1) The authors suggest based on comparative lipidomics analysis of SMALPs and of mitochondrial membranes that the extraction of lipids from the MOM is inadequate for cardiolipin (CL), sphingomyelins, and ceramides. Would it be possible that the lack of CL, SM, and ceramides in SMALPS is due to exclusion from the nano-environment of BAK? Also, some ceramides appeared at a substantial/comparable fraction within the SMALPs (Supp.Fig. 1E), i.e. the approach appears to be suitable also for (at least some) ceramides.

2) Is it known / can it be determined how many BAKs are found on average within a lipid nanodisc? I.e. do they form clusters? Could this be addressed in additional simulations?

3) The authors conclude from analysis of liposome permeabilization that unsaturated lipids promote membrane permeabilization (p6, Fig. 3A-C). Since DPPC was likely in the gel phase this study does hardly allow for comparison of the effect of saturated with unsaturated lipids. Could this experiment as well be performed at sufficiently high temperature, i.e. within the fluid phase of DPPC? The authors may want to discuss a previous permeabilization study focusing on the role of lipid asymmetry (Bozelli et al., BBA-Biomembranes 1862, 2020, 183241).

4) For the GUV experiments, which ratio of POPC to (h)CL was employed? (Fig.3E,F)

5) MD simulations: Was the membrane modeled as a symmetric lipid bilayer? What is known about lipid asymmetry of MOM? The authors observed enrichment of PC lipids in pore vicinity, despite an assumed cylindrical shape (how was vicinity defined?). A recent MD paper indeed showed that also PC lipids with unsaturated chains have a negative spontaneous curvature (Pohnl et al., JCTC 19, 2023, 1908).

6) MD: The PMF profile for pore formation lacks an error analysis. This should be added.

7) A main result of the manuscript is that unsaturated lipids preferentially partition to BAX/BAK. It would be great if this result from lipidomics and SMALPs could be underpinned with simulations. It was shown before for other proteins (e.g. GPCRs) that a number of proteins favor unsaturated acyl chains in their vicinity.

Reviewer #2 (Remarks to the Author):

Attached

The action of Bcl-2 proteins, such as pro-apoptotic BAX and BAK, is intimately associated with their interaction with membranes. Consequently, the properties of the lipid bilayer are expected to influence such activities profoundly. Despite this expectation, the subject of the modulation of the BAX and BAK activity by the physicochemical properties of the bilayer received inadequate attention. This manuscript addresses the gap in our knowledge on the subject by presenting overwhelming evidence that lipid unsaturation promotes pore-forming activity of BAX and BAK. The authors establish this finding using a variety of systems, from cells and mitochondria, to model vesicles with well-controlled lipid compositions, to Coarse-grain molecular dynamics computer simulations. They also reports on the enrichment of polyunsaturated lipids in apoptotic mitochondria and in the vicinity of the apoptotic pore. And rightfully conclude that increased lipid unsaturation has functional consequences for apoptosis progression by promoting the BAX and BAK pore activity and, consequently, cell death sensitivity.

All experimental and computational work is designed and performed carefully completed with the appropriate controls and using a number of complementary approaches, all conveying the same message of the importance of lipid unsaturation. I have only a minor comment about the conditions for the experiments with model membranes. It is not clear if the were performed in the presence of Mg^{2+} ions. The reason for this comes from a recently published study that identified Mg^{2+} ions as coupling agents between Bcl-2 proteins (BAX included) and lipids. This study is also relevant to the BH-3 independent activation and targeting of BAX and other Bcl-2 proteins, discussed in the introduction.

Reviewer #3 (Remarks to the Author):

In this manuscript, Dadsena and colleagues explore the contribution of lipids to BAK regulation and the opening of the mitochondria PTP. Using a SMAPL -approach, the authors report that membranes attached to recombinant BAK are rich in unsaturated lipids, and in particular, after apoptotic. Based on their data, they propose that unsaturated lipids promote BAX pore activity, and that FADS2 correlate with increased sensitivity to apoptosis.

While this work has many merits, I believe there are major weaknesses that preclude its publication at this stage. In particular, the lack of appropriate controls impedes the interpretations of the data presented. Also, some technical concerns need to be resolved to supports the authors' conclusions.

1) SMA is a novel approach that allows for the study of membrane proteins in their environment, however it does not solubilize all kinds of lipids equally. Previous studies indicate that, while not showing a preference for any specific class, SMALPs is less efficient solubilizing lipid species bound to saturated fatty acids. This characteristic could help clarify why the authors observed a clear enrichment in unsaturated phospholipid species, and why they were not able to solubilize ceramide or sphingomyelin, as the large bulk of sphingolipid species are bound to saturated fatty acids. In fact, this is illustrated in their supplementary figure 1, where they show that SMA treatment has in fact a significant effect in the solubilization of saturated species, some of which are quite abundant in mitochondrial membranes (e.g., PC 32.0). The analysis of the material not solubilized by SMALPs could help clarify these points. This point is essential in their interpretations, since the regulation of the mPTP has been shown to be highly sensitive to cholesterol and sphingolipid content, and ceramides in particular.

2) Heatmaps in Fig. 2A show a clear inconsistency among replicates that lowers the significance of these data.

3) It is unclear if the ablation of Bax has an effect on the lipid environment of Bak.

4) Their mitochondrial preparation is not pure and probably contaminated with many membrane organelles, not only mitochondria.

5) Fig. 3I. It is unclear the role of Bax in the internalization of both dyes. It is already well-known that membrane unsaturation increases its permeability.

6) In Fig. 5. Data is difficult to interpret. It is unclear how they add the FA to their cultures and whether these lipids are incorporated into their cell models.

7) Fig 6. As before, the authors must show that in their cultures, the added FA is indeed incorporated into cellular membranes. In Fig. 6A. The KO shows FADS2 expression. The authors need to control for the expression of other FA desaturases and CoA-ligases. Also, they need to show that mitochondria from FADSKO cells have indeed more saturated membranes. This point is rather important, as this data is opposed to many other reports showing decreased apoptotic rates after incubation with polyunsaturated FA.

RESPONSE TO REVIEWERS' COMMENT

We are grateful to the reviewers for the positive evaluation of our work and for their helpful suggestions.

Reviewer #1

The manuscript by Dadsena et al. focuses on the role of lipid unsaturation near BAX and BAK for mitochondrial outer membrane permeabilization (MOMP). Combining analysis of SMA stabilized BAK-containing lipid nanodiscs with molecular dynamics (MD) simulations they show that BAX/BAK prefer unsaturated lipids in their nano-environment and that this lipid environment may promote membrane permeabilization. Overall a very nice study!

The following points need some clarification:

1) The authors suggest based on comparative lipidomics analysis of SMALPs and of mitochondrial membranes that the extraction of lipids from the MOM is inadequate for cardiolipin (CL), sphingomyelins, and ceramides. Would it be possible that the lack of CL, SM, and ceramides in SMALPS is due to exclusion from the nano-environment of BAK? Also, some ceramides appeared at a substantial/comparable fraction within the SMALPs (Supp.Fig. 1E), i.e. the approach appears to be suitable also for (at least some) ceramides.

We thank the reviewer for raising this question. As mentioned in the manuscript, we do not see any traces of CL within the mEGFP-BAK SMALPs, but indeed we found little amounts of SM and Ceramides, which we now include as Supplementary Fig 1F and discuss in the manuscript. However, since we cannot detect with confidence good levels of CL, sphingomyelins and ceramides in total SMALPs samples, we did not want to make conclusions about these lipid species being excluded from the enriched mEGFP-BAK SMALPS.

2) Is it known / can it be determined how many BAKs are found on average within a lipid nanodisc? I.e. do they form clusters? Could this be addressed in additional simulations?

Currently we do not know the number of BAK molecules per nanodisc, although we know that BAK is not clustered when solubilized with SMALPs from healthy cells, and that it is clustered when we solubilize it with SMALPS from apoptotic cells. In Cosentino et al. Mol Cell, 2022, we investigated the copy number of BAK per cluster during apoptosis, which was dynamic and broadly distributed ranging from a few protein to copies to a few hundreds. We expect that the solubilization with SMALPs will break apart large BAK clusters in apoptotic conditions because they are larger than the size of the SMALPS or that only the cluster size that fits into the SMALPs will be incorporated.

Regarding the simulations, unfortunately, simulating such a system would be fraught with uncertainty (Pourmoussa & Pastor, 2018) even if assumed we could accurately recover the BAK-BAK oligomeric configurations (see also response to question 7). We can only envision to address this question using single molecule stoichiometry analysis of GFP-BAK in isolated SMALPs measured with TIRF microscopy. While we should be technically capable of performing this analysis, it would require a large experimental effort and time. We do not see that determining the stoichiometry of BAK molecules in

the SMALPs adds to the findings reported in our study, since it will not reflect the size of the clusters in the cell, which we already characterized in the past.

3) The authors conclude from analysis of liposome permeabilization that unsaturated lipids promote membrane permeabilization (p6, Fig. 3A-C). Since DPPC was likely in the gel phase this study does hardly allow for comparison of the effect of saturated with unsaturated lipids. Could this experiment as well be performed at sufficiently high temperature, i.e. within the fluid phase of DPPC? The authors may want to discuss a previous permeabilization study focusing on the role of lipid asymmetry (Bozelli et al., BBA-Biomembranes 1862, 2020, 183241).

This is a good point that we also considered during our study. However, Bax becomes activated with mild temperatures, which precludes us from performing this experiment at a higher temperature. For this reason, we decided to use lipid mixtures with varying levels of lipid unsaturation, as shown in Figure 1 for LUVs and GUVs. We kept the fully saturated mixture made of DPPC:CL(16:0) as a reference, but discuss in the manuscript that the lack of activity is most likely due to the membrane being in the gel phase.

During the revisions, we tried to address this question by performing the liposome permeabilization assay with another saturated PC lipid with a lower T_m (DMPC; 24°C). However, liposomes with such saturated lipid composition also remained in the gel phase and precluded performing these experiments.

4) For the GUV experiments, which ratio of POPC to (h)CL was employed? (Fig.3E,F)

We used PC to CL or (h)CL always in 8:2 ratio. We mentioned this in the main text in addition to the methods section.

5) MD simulations: Was the membrane modeled as a symmetric lipid bilayer? What is known about lipid asymmetry of MOM? The authors observed enrichment of PC lipids in pore vicinity, despite an assumed cylindrical shape (how was vicinity defined?). A recent MD paper indeed showed that also PC lipids with unsaturated chains have a negative spontaneous curvature (Pohnl et al., JCTC 19, 2023, 1908).

Thank you for raising this point. The MOM was modeled with asymmetric lipid distributions as follows: the lipidomics assays were used to determine the overall lipid types and their proportions. These were then divided into cytoplasmic/intermembrane-space leaflets assuming the proportions in Fig. 1B of Horvath & Daum 2013 (Horvath & Daum, 2013) (30/70, 30/70, 77/23, and 55/45 for PS, PI, PE, and PC, respectively; information on PG, which is a minority component, was not available and 50/50 distribution was assumed). In order to avoid introducing tension due to area-per-lipid mismatch between leaflets, each composition's area-per-lipid was determined from symmetric simulations. Leaflets were then juxtaposed, after proportionally removing enough lipids from the one with highest area-per-lipid, so that expected areas matched. This information has been added to the Methods.

Pore vicinity was defined as a 4-nm cylindrical zone centered on the pore. Any lipid with a particle inside this cylinder was considered to be pore-associated. This has been better detailed in the methods section. We also discuss the work remarked by the reviewer in the main text.

6) MD: The PMF profile for pore formation lacks an error analysis. This should be added.

The estimated error had been calculated as per the bootstrap method of the WHAM software (now explicitly stated in the methods). It is, however, of magnitude between 10^{-1} – 10^0 kJ/mol and thus not visible in the 10^2 kJ/mol scale of the PMF profiles. This is now remarked in the caption.

7) *A main result of the manuscript is that unsaturated lipids preferentially partition to BAX/BAK. It would be great if this result from lipidomics and SMALPs could be underpinned with simulations. It was shown before for other proteins (e.g. GPCRs) that a number of proteins favor unsaturated acyl chains in their vicinity.*

This is a very good point, which we also considered and would love to address. However, available structures of active BAX or BAK including membrane-anchored domains are lacking, which precludes the use of CG MD such as the Martini model, which cannot evolve secondary structure on their own. The use of atomistic MD could circumvent, at least in part, this limitation. However, in that case folding and lipid dynamics are likely prohibitively slow, computation-wise, for converged statistics to be accumulated.

Reviewer #2

The action of Bcl-2 proteins, such as pro-apoptotic BAX and BAK, is intimately associated with their interaction with membranes. Consequently, the properties of the lipid bilayer are expected to influence such activities profoundly. Despite this expectation, the subject of the modulation of the BAX and BAK activity by the physicochemical properties of the bilayer received inadequate attention. This manuscript addresses the gap in our knowledge on the subject by presenting overwhelming evidence that lipid unsaturation promotes pore-forming activity of BAX and BAK. The authors establish this finding using a variety of systems, from cells and mitochondria, to model vesicles with well-controlled lipid compositions, to Coarse-grain molecular dynamics computer simulations. They also report on the enrichment of polyunsaturated lipids in apoptotic mitochondria and in the vicinity of the apoptotic pore. And rightfully conclude that increased lipid unsaturation has functional consequences for apoptosis progression by promoting the BAX and BAK pore activity and, consequently, cell death sensitivity.

All experimental and computational work is designed and performed carefully completed with the appropriate controls and using a number of complementary approaches, all conveying the same message of the importance of lipid unsaturation. I have only a minor comment about the conditions for the experiments with model membranes. It is not clear if the were performed in the presence of Mg²⁺ ions. The reason for this comes from a recently published study that identified Mg²⁺ ions as coupling agents between Bcl-2 proteins (BAX included) and lipids. This study is also relevant to the BH-3 independent activation and targeting of BAX and other Bcl-2 proteins, discussed in the introduction.

This is an interesting aspect. Indeed, membrane insertion of BCL2 proteins is affected by divalent cations such as Ca²⁺ or Mg²⁺, as reported by Vasquez-Montes et al. 2021, which we now included in

the introduction of the revised manuscript. However, our model membrane experiments on GUVs and liposomes are free of any divalent cation.

Reviewer #3

In this manuscript, Dadsena and colleagues explore the contribution of lipids to BAK regulation and the opening of the mitochondria PTP. Using a SMALP -approach, the authors report that membranes attached to recombinant BAK are rich in unsaturated lipids, and in particular, after apoptotic. Based on their data, they propose that unsaturated lipids promote BAX pore activity, and that FADS2 correlate with increased sensitivity to apoptosis. While this work has many merits, I believe there are major weaknesses that preclude its publication at this stage. In particular, the lack of appropriate controls impedes the interpretations of the data presented. Also, some technical concerns need to be resolved to supports the authors' conclusions.

We thank the reviewer for the positive assessment of our study. We would like to point out that all our experiments have been performed under conditions of apoptosis induction dependent on specific activation of BAX/BAK and subsequent mitochondrial outer membrane permeabilization, also known as MOMP, leading to caspase dependent cell death, as supported by our previous work and an extensive body of literature. Our study does not include experimental conditions inducing opening of the mitochondrial PTP, which leads to a form of regulated necrosis different from apoptosis.

1) SMA is a novel approach that allows for the study of membrane proteins in their environment, however it does not solubilize all kinds of lipids equally. Previous studies indicate that, while not showing a preference for any specific class, SMALPs is less efficient solubilizing lipid species bound to saturated fatty acids. This characteristic could help clarify why the authors observed a clear enrichment in unsaturated phospholipid species, and why they were not able to solubilize ceramide or sphingomyelin, as the large bulk of sphingolipid species are bound to saturated fatty acids. In fact, this is illustrated in their supplementary figure 1, where they show that SMA treatment has in fact a significant effect in the solubilization of saturated species, some of which are quite abundant in mitochondrial membranes (e.g., PC 32.0). The analysis of the material not solubilized by SMALPs could help clarify these points. This point is essential in their interpretations, since the regulation of the mPTP has been shown to be highly sensitive to cholesterol and sphingolipid content, and ceramides in particular.

We agree with the reviewer that the differential efficiency of SMALPs in the solubilization of lipid species is a factor that needs to be taken in to account, which we did in our study. It is for this reason that we also analyzed the lipidome of total mitochondrial extracts from healthy and apoptotic cells and compared it to the lipidome of total SMALPs extracted under the same conditions. By comparing mitochondria control vs. SMALPs control we could identify those species that are present in mitochondria but not efficiently permeabilized into SMALPs, such as cardiolipins, sphingomyelins and ceramides, also in agreement with the literature. As the reviewer suggests, saturated lipids are also less soluble in SMA in the case of PC species (shown in Supplementary Fig. 1A), while this is not so obvious for PE, PI and PG (Supplementary figure 1B-1D). We refrained from further analyzing lipids not

efficiently included in SMALPs and from making conclusions about these lipids. We focused instead in the lipids that are efficiently incorporated into SMALPs, which include the unsaturated lipid species.

Once this point was clear, we then analyzed the lipids that changed in mEGFP-BAK-enriched SMALPs extracted from apoptotic cells compared to healthy cells. The changes detected here correspond to changes in the SMALPs lipidome due to the induction of apoptosis and not to the efficiency of extraction, since we are comparing the same lipids from apoptotic and control healthy cells and thus their solubilization efficiency is the same. As a result, the concern raised by the reviewer does not apply.

In our study, we have not investigated the lipid effects on mPTP but on the BAX/BAK pores induced during MOMP. There are some reports regarding the role of cholesterol and sphingolipid derivatives on MOMP, but specially cardiolipin has been proposed to have more relevance. Unfortunately, our experimental approach is not optimal to study these lipids as discussed above, yet, we have now included the data for eGFP-BAK enriched SMALPs changes in sphingomyelins and ceramides in Supplementary Figure 1F.

2) Heatmaps in Fig. 2A show a clear inconsistency among replicates that lowers the significance of these data.

We agree that there is variability in the data obtained, but we think this was hard to avoid considering the heterogeneous and dynamic system (cells undergoing apoptosis) that we have investigated. For this reason, we performed four replicates instead of three. In the revised version, we made heatmap graphs in Figure 2B by using raw value (nmol/mg protein) instead of mol % values for more accurate presentation. In addition, we avoided making conclusions from this analysis on individual lipid species and focused instead on general trends, which we later characterized in several experimental systems to support our conclusions.

3) It is unclear if the ablation of Bax has an effect on the lipid environment of Bak.

We have performed our lipidomics experiments on U2OS BAK KO cells stably transfected with mEGFP-BAK, where endogenous BAX is present. We also checked in previous work that purified mEGFP-BAK SMALPs also contain endogenous BAX (see Cosentino et al, Mol Cell, 2022), which we mention in the text.

4) Their mitochondrial preparation is not pure and probably contaminated with many membrane organelles, not only mitochondria.

Thank you for raising this point. As the reviewer mentions, our mitochondrial preparation is not pure, but “crude”, which we mention more explicitly now in the manuscript. It is practically impossible to prepare pure mitochondria because small amounts of organelles co-purify with mitochondria as a result of inter-organelle contacts in the cell, which is especially relevant in the case of the endoplasmic reticulum. We have used here protocols for mitochondria isolation that are standard in the apoptosis field to investigate pore formation by BAX and BAK. Yet, we find the reviewer’s comment something that readers should be aware of. To this aim, in the revised manuscript we have included a new panel

with the analysis of the purity of our mitochondrial preparations by western blot using markers for different cellular organelles (Supplementary Figure 2A) and discussed this aspect in the text.

5) Fig. 3I. It is unclear the role of Bax in the internalization of both dyes. It is already well-known that membrane unsaturation increases its permeability.

The role of BAX is to create the pores that permeabilize the vesicles. This assay estimates the size of BAX pores by determining the ability of the dyes used in the assay, which have different molecular weight (cyt c is 12kDa, APC is about 100kDa), to enter the individual GUVs. In absence of BAX the vesicles are not significantly permeabilized, as we show in the negative controls of this graph. Membrane unsaturation alone does not permeabilize membranes under the conditions used in our experiments.

6) In Fig. 5. Data is difficult to interpret. It is unclear how they add the FA to their cultures and whether these lipids are incorporated into their cell models.

We thank the reviewer for this question. Fatty acid treatment to the cells was done by adding ethanolic stock of fatty acids into culture media using an established method as described in (Haberkant et al., 2013). In the revised version of the manuscript, we have added this information in the methods section. We also confirmed now that they are incorporated into mitochondria, which are the membrane systems of relevance in our study. To this aim, we have performed a lipidomic analysis of total fatty acids in mitochondrial membrane isolated from U2OS WT and FADS2KO cells treated or not with linoleic acid (LA;18:2). As shown in the new Supplementary Figure 4, we could detect the incorporation of LA into mitochondrial membranes in both cell lines. In addition, in wt but not FADS2KO cells, LA was further converted into polyunsaturated fatty acids, as shown by the increase of 18:3, 20:3 and 20:4 levels (Supplementary Figure 4).

7) Fig 6. As before, the authors must show that in their cultures, the added FA is indeed incorporated into cellular membranes.

To keep costs reasonable, we have performed the requested lipidomic analysis in selected cell lines that are representative for our study, as mentioned in the response to point 6). The results obtained (shown in Supplementary Figure 4) indicate that our protocol efficiently incorporates the added FA to mitochondrial membranes.

In Fig. 6A. The KO shows FADS2 expression. The authors need to control for the expression of other FA desaturases and CoA-ligases.

Thank you for this suggestion. In the revised manuscript, we have now evaluated the expression levels of FADS1 and ACSL4, as examples of other FA desaturases and CoA-ligases, in the different cell lines used in the study using western blot. The levels of FADS1 do not significantly change between the lung cancer cells, and they seem slightly lower in the OS-RC2 kidney cancer cells, in good agreement with the cell death results obtained. With respect to ACSL4, there seems to be no correlation between

expression levels and cell death sensitivity. These data are now included in the revised manuscript in Supplementary Figure 2B.

Also, they need to show that mitochondria from FADSKO cells have indeed more saturated membranes. This point is rather important, as this data is opposed to many other reports showing decreased apoptotic rates after incubation with polyunsaturated FA.

This point is also addressed in the same experiment designed to answer question 6 and the first part of question 7. As shown in Supplementary Figure 4, the mitochondria of FADS2 KO cells contain lower levels of polyunsaturated fatty acids, including 20:3, 20:4 and 22:6.

REVIEWERS' COMMENTS

Reviewer #1 (Remarks to the Author):

The authors have thoroughly and effectively addressed each point raised, providing detailed explanations and evidence where necessary.

Reviewer #2 (Remarks to the Author):

My comment has been addressed

Reviewer #3 (Remarks to the Author):

I thank the authors for addressing each of my comments. The manuscript has undergone significant improvement; however, there remain some points of ambiguity concerning the variability in some of the presented assays and/or the role of other desaturases on FADSKO cells. Despite these uncertainties, I recognize that such limitations are inherent to the methodology. The authors have thoughtfully discussed these aspects, and I believe they should not hinder the publication of this interesting work

RESPONSE TO REVIEWERS' COMMENT

We are grateful to the reviewers for the positive evaluation of our work and for their helpful suggestions.

Reviewer #1

The manuscript by Dadsena et al. focuses on the role of lipid unsaturation near BAX and BAK for mitochondrial outer membrane permeabilization (MOMP). Combining analysis of SMA stabilized BAK-containing lipid nanodiscs with molecular dynamics (MD) simulations they show that BAX/BAK prefer unsaturated lipids in their nano-environment and that this lipid environment may promote membrane permeabilization. Overall a very nice study!

The following points need some clarification:

1) The authors suggest based on comparative lipidomics analysis of SMALPs and of mitochondrial membranes that the extraction of lipids from the MOM is inadequate for cardiolipin (CL), sphingomyelins, and ceramides. Would it be possible that the lack of CL, SM, and ceramides in SMALPS is due to exclusion from the nano-environment of BAK? Also, some ceramides appeared at a substantial/comparable fraction within the SMALPs (Supp.Fig. 1E), i.e. the approach appears to be suitable also for (at least some) ceramides.

We thank the reviewer for raising this question. As mentioned in the manuscript, we do not see any traces of CL within the mEGFP-BAK SMALPs, but indeed we found little amounts of SM and Ceramides, which we now include as Supplementary Fig 1F and discuss in the manuscript. However, since we cannot detect with confidence good levels of CL, sphingomyelins and ceramides in total SMALPs samples, we did not want to make conclusions about these lipid species being excluded from the enriched mEGFP-BAK SMALPS.

2) Is it known / can it be determined how many BAKs are found on average within a lipid nanodisc? I.e. do they form clusters? Could this be addressed in additional simulations?

Currently we do not know the number of BAK molecules per nanodisc, although we know that BAK is not clustered when solubilized with SMALPs from healthy cells, and that it is clustered when we solubilize it with SMALPS from apoptotic cells. In Cosentino et al. Mol Cell, 2022, we investigated the copy number of BAK per cluster during apoptosis, which was dynamic and broadly distributed ranging from a few protein to copies to a few hundreds. We expect that the solubilization with SMALPs will break apart large BAK clusters in apoptotic conditions because they are larger than the size of the SMALPS or that only the cluster size that fits into the SMALPs will be incorporated.

Regarding the simulations, unfortunately, simulating such a system would be fraught with uncertainty (Pourmoussa & Pastor, 2018) even if assumed we could accurately recover the BAK-BAK oligomeric configurations (see also response to question 7). We can only envision to address this question using single molecule stoichiometry analysis of GFP-BAK in isolated SMALPs measured with TIRF microscopy. While we should be technically capable of performing this analysis, it would require a large experimental effort and time. We do not see that determining the stoichiometry of BAK molecules in

the SMALPs adds to the findings reported in our study, since it will not reflect the size of the clusters in the cell, which we already characterized in the past.

3) The authors conclude from analysis of liposome permeabilization that unsaturated lipids promote membrane permeabilization (p6, Fig. 3A-C). Since DPPC was likely in the gel phase this study does hardly allow for comparison of the effect of saturated with unsaturated lipids. Could this experiment as well be performed at sufficiently high temperature, i.e. within the fluid phase of DPPC? The authors may want to discuss a previous permeabilization study focusing on the role of lipid asymmetry (Bozelli et al., BBA-Biomembranes 1862, 2020, 183241).

This is a good point that we also considered during our study. However, Bax becomes activated with mild temperatures, which precludes us from performing this experiment at a higher temperature. For this reason, we decided to use lipid mixtures with varying levels of lipid unsaturation, as shown in Figure 1 for LUVs and GUVs. We kept the fully saturated mixture made of DPPC:CL(16:0) as a reference, but discuss in the manuscript that the lack of activity is most likely due to the membrane being in the gel phase.

During the revisions, we tried to address this question by performing the liposome permeabilization assay with another saturated PC lipid with a lower T_m (DMPC; 24°C). However, liposomes with such saturated lipid composition also remained in the gel phase and precluded performing these experiments.

4) For the GUV experiments, which ratio of POPC to (h)CL was employed? (Fig.3E,F)

We used PC to CL or (h)CL always in 8:2 ratio. We mentioned this in the main text in addition to the methods section.

5) MD simulations: Was the membrane modeled as a symmetric lipid bilayer? What is known about lipid asymmetry of MOM? The authors observed enrichment of PC lipids in pore vicinity, despite an assumed cylindrical shape (how was vicinity defined?). A recent MD paper indeed showed that also PC lipids with unsaturated chains have a negative spontaneous curvature (Pohnl et al., JCTC 19, 2023, 1908).

Thank you for raising this point. The MOM was modeled with asymmetric lipid distributions as follows: the lipidomics assays were used to determine the overall lipid types and their proportions. These were then divided into cytoplasmic/intermembrane-space leaflets assuming the proportions in Fig. 1B of Horvath & Daum 2013 (Horvath & Daum, 2013) (30/70, 30/70, 77/23, and 55/45 for PS, PI, PE, and PC, respectively; information on PG, which is a minority component, was not available and 50/50 distribution was assumed). In order to avoid introducing tension due to area-per-lipid mismatch between leaflets, each composition's area-per-lipid was determined from symmetric simulations. Leaflets were then juxtaposed, after proportionally removing enough lipids from the one with highest area-per-lipid, so that expected areas matched. This information has been added to the Methods.

Pore vicinity was defined as a 4-nm cylindrical zone centered on the pore. Any lipid with a particle inside this cylinder was considered to be pore-associated. This has been better detailed in the methods section. We also discuss the work remarked by the reviewer in the main text.

6) MD: The PMF profile for pore formation lacks an error analysis. This should be added.

The estimated error had been calculated as per the bootstrap method of the WHAM software (now explicitly stated in the methods). It is, however, of magnitude between 10^{-1} – 10^0 kJ/mol and thus not visible in the 10^2 kJ/mol scale of the PMF profiles. This is now remarked in the caption.

7) *A main result of the manuscript is that unsaturated lipids preferentially partition to BAX/BAK. It would be great if this result from lipidomics and SMALPs could be underpinned with simulations. It was shown before for other proteins (e.g. GPCRs) that a number of proteins favor unsaturated acyl chains in their vicinity.*

This is a very good point, which we also considered and would love to address. However, available structures of active BAX or BAK including membrane-anchored domains are lacking, which precludes the use of CG MD such as the Martini model, which cannot evolve secondary structure on their own. The use of atomistic MD could circumvent, at least in part, this limitation. However, in that case folding and lipid dynamics are likely prohibitively slow, computation-wise, for converged statistics to be accumulated.

Reviewer #2

The action of Bcl-2 proteins, such as pro-apoptotic BAX and BAK, is intimately associated with their interaction with membranes. Consequently, the properties of the lipid bilayer are expected to influence such activities profoundly. Despite this expectation, the subject of the modulation of the BAX and BAK activity by the physicochemical properties of the bilayer received inadequate attention. This manuscript addresses the gap in our knowledge on the subject by presenting overwhelming evidence that lipid unsaturation promotes pore-forming activity of BAX and BAK. The authors establish this finding using a variety of systems, from cells and mitochondria, to model vesicles with well-controlled lipid compositions, to Coarse-grain molecular dynamics computer simulations. They also report on the enrichment of polyunsaturated lipids in apoptotic mitochondria and in the vicinity of the apoptotic pore. And rightfully conclude that increased lipid unsaturation has functional consequences for apoptosis progression by promoting the BAX and BAK pore activity and, consequently, cell death sensitivity.

All experimental and computational work is designed and performed carefully completed with the appropriate controls and using a number of complementary approaches, all conveying the same message of the importance of lipid unsaturation. I have only a minor comment about the conditions for the experiments with model membranes. It is not clear if the were performed in the presence of Mg²⁺ ions. The reason for this comes from a recently published study that identified Mg²⁺ ions as coupling agents between Bcl-2 proteins (BAX included) and lipids. This study is also relevant to the BH-3 independent activation and targeting of BAX and other Bcl-2 proteins, discussed in the introduction.

This is an interesting aspect. Indeed, membrane insertion of BCL2 proteins is affected by divalent cations such as Ca²⁺ or Mg²⁺, as reported by Vasquez-Montes et al. 2021, which we now included in

the introduction of the revised manuscript. However, our model membrane experiments on GUVs and liposomes are free of any divalent cation.

Reviewer #3

In this manuscript, Dadsena and colleagues explore the contribution of lipids to BAK regulation and the opening of the mitochondria PTP. Using a SMALP -approach, the authors report that membranes attached to recombinant BAK are rich in unsaturated lipids, and in particular, after apoptotic. Based on their data, they propose that unsaturated lipids promote BAX pore activity, and that FADS2 correlate with increased sensitivity to apoptosis. While this work has many merits, I believe there are major weaknesses that preclude its publication at this stage. In particular, the lack of appropriate controls impedes the interpretations of the data presented. Also, some technical concerns need to be resolved to supports the authors' conclusions.

We thank the reviewer for the positive assessment of our study. We would like to point out that all our experiments have been performed under conditions of apoptosis induction dependent on specific activation of BAX/BAK and subsequent mitochondrial outer membrane permeabilization, also known as MOMP, leading to caspase dependent cell death, as supported by our previous work and an extensive body of literature. Our study does not include experimental conditions inducing opening of the mitochondrial PTP, which leads to a form of regulated necrosis different from apoptosis.

1) SMA is a novel approach that allows for the study of membrane proteins in their environment, however it does not solubilize all kinds of lipids equally. Previous studies indicate that, while not showing a preference for any specific class, SMALPs is less efficient solubilizing lipid species bound to saturated fatty acids. This characteristic could help clarify why the authors observed a clear enrichment in unsaturated phospholipid species, and why they were not able to solubilize ceramide or sphingomyelin, as the large bulk of sphingolipid species are bound to saturated fatty acids. In fact, this is illustrated in their supplementary figure 1, where they show that SMA treatment has in fact a significant effect in the solubilization of saturated species, some of which are quite abundant in mitochondrial membranes (e.g., PC 32.0). The analysis of the material not solubilized by SMALPs could help clarify these points. This point is essential in their interpretations, since the regulation of the mPTP has been shown to be highly sensitive to cholesterol and sphingolipid content, and ceramides in particular.

We agree with the reviewer that the differential efficiency of SMALPs in the solubilization of lipid species is a factor that needs to be taken in to account, which we did in our study. It is for this reason that we also analyzed the lipidome of total mitochondrial extracts from healthy and apoptotic cells and compared it to the lipidome of total SMALPs extracted under the same conditions. By comparing mitochondria control vs. SMALPs control we could identify those species that are present in mitochondria but not efficiently permeabilized into SMALPs, such as cardiolipins, sphingomyelins and ceramides, also in agreement with the literature. As the reviewer suggests, saturated lipids are also less soluble in SMA in the case of PC species (shown in Supplementary Fig. 1A), while this is not so obvious for PE, PI and PG (Supplementary figure 1B-1D). We refrained from further analyzing lipids not

efficiently included in SMALPs and from making conclusions about these lipids. We focused instead in the lipids that are efficiently incorporated into SMALPs, which include the unsaturated lipid species.

Once this point was clear, we then analyzed the lipids that changed in mEGFP-BAK-enriched SMALPs extracted from apoptotic cells compared to healthy cells. The changes detected here correspond to changes in the SMALPs lipidome due to the induction of apoptosis and not to the efficiency of extraction, since we are comparing the same lipids from apoptotic and control healthy cells and thus their solubilization efficiency is the same. As a result, the concern raised by the reviewer does not apply.

In our study, we have not investigated the lipid effects on mPTP but on the BAX/BAK pores induced during MOMP. There are some reports regarding the role of cholesterol and sphingolipid derivatives on MOMP, but specially cardiolipin has been proposed to have more relevance. Unfortunately, our experimental approach is not optimal to study these lipids as discussed above, yet, we have now included the data for eGFP-BAK enriched SMALPs changes in sphingomyelins and ceramides in Supplementary Figure 1F.

2) Heatmaps in Fig. 2A show a clear inconsistency among replicates that lowers the significance of these data.

We agree that there is variability in the data obtained, but we think this was hard to avoid considering the heterogeneous and dynamic system (cells undergoing apoptosis) that we have investigated. For this reason, we performed four replicates instead of three. In the revised version, we made heatmap graphs in Figure 2B by using raw value (nmol/mg protein) instead of mol % values for more accurate presentation. In addition, we avoided making conclusions from this analysis on individual lipid species and focused instead on general trends, which we later characterized in several experimental systems to support our conclusions.

3) It is unclear if the ablation of Bax has an effect on the lipid environment of Bak.

We have performed our lipidomics experiments on U2OS BAK KO cells stably transfected with mEGFP-BAK, where endogenous BAX is present. We also checked in previous work that purified mEGFP-BAK SMALPs also contain endogenous BAX (see Cosentino et al, Mol Cell, 2022), which we mention in the text.

4) Their mitochondrial preparation is not pure and probably contaminated with many membrane organelles, not only mitochondria.

Thank you for raising this point. As the reviewer mentions, our mitochondrial preparation is not pure, but “crude”, which we mention more explicitly now in the manuscript. It is practically impossible to prepare pure mitochondria because small amounts of organelles co-purify with mitochondria as a result of inter-organelle contacts in the cell, which is especially relevant in the case of the endoplasmic reticulum. We have used here protocols for mitochondria isolation that are standard in the apoptosis field to investigate pore formation by BAX and BAK. Yet, we find the reviewer’s comment something that readers should be aware of. To this aim, in the revised manuscript we have included a new panel

with the analysis of the purity of our mitochondrial preparations by western blot using markers for different cellular organelles (Supplementary Figure 2A) and discussed this aspect in the text.

5) Fig. 3I. It is unclear the role of Bax in the internalization of both dyes. It is already well-known that membrane unsaturation increases its permeability.

The role of BAX is to create the pores that permeabilize the vesicles. This assay estimates the size of BAX pores by determining the ability of the dyes used in the assay, which have different molecular weight (cyt c is 12kDa, APC is about 100kDa), to enter the individual GUVs. In absence of BAX the vesicles are not significantly permeabilized, as we show in the negative controls of this graph. Membrane unsaturation alone does not permeabilize membranes under the conditions used in our experiments.

6) In Fig. 5. Data is difficult to interpret. It is unclear how they add the FA to their cultures and whether these lipids are incorporated into their cell models.

We thank the reviewer for this question. Fatty acid treatment to the cells was done by adding ethanolic stock of fatty acids into culture media using an established method as described in (Haberkant et al., 2013). In the revised version of the manuscript, we have added this information in the methods section. We also confirmed now that they are incorporated into mitochondria, which are the membrane systems of relevance in our study. To this aim, we have performed a lipidomic analysis of total fatty acids in mitochondrial membrane isolated from U2OS WT and FADS2KO cells treated or not with linoleic acid (LA;18:2). As shown in the new Supplementary Figure 4, we could detect the incorporation of LA into mitochondrial membranes in both cell lines. In addition, in wt but not FADS2KO cells, LA was further converted into polyunsaturated fatty acids, as shown by the increase of 18:3, 20:3 and 20:4 levels (Supplementary Figure 4).

7) Fig 6. As before, the authors must show that in their cultures, the added FA is indeed incorporated into cellular membranes.

To keep costs reasonable, we have performed the requested lipidomic analysis in selected cell lines that are representative for our study, as mentioned in the response to point 6). The results obtained (shown in Supplementary Figure 4) indicate that our protocol efficiently incorporates the added FA to mitochondrial membranes.

In Fig. 6A. The KO shows FADS2 expression. The authors need to control for the expression of other FA desaturases and CoA-ligases.

Thank you for this suggestion. In the revised manuscript, we have now evaluated the expression levels of FADS1 and ACSL4, as examples of other FA desaturases and CoA-ligases, in the different cell lines used in the study using western blot. The levels of FADS1 do not significantly change between the lung cancer cells, and they seem slightly lower in the OS-RC2 kidney cancer cells, in good agreement with the cell death results obtained. With respect to ACSL4, there seems to be no correlation between

expression levels and cell death sensitivity. These data are now included in the revised manuscript in Supplementary Figure 2B.

Also, they need to show that mitochondria from FADSKO cells have indeed more saturated membranes. This point is rather important, as this data is opposed to many other reports showing decreased apoptotic rates after incubation with polyunsaturated FA.

This point is also addressed in the same experiment designed to answer question 6 and the first part of question 7. As shown in Supplementary Figure 4, the mitochondria of FADS2 KO cells contain lower levels of polyunsaturated fatty acids, including 20:3, 20:4 and 22:6.